# Ripply suppresses Tbx6 to induce dynamic-to-static conversion in somite segmentation

Taijiro Yabe [1,2,3] ✉, Koichiro Uriu [4] ✉ & Shinji Takada [1,2,3] ✉

The metameric pattern of somites is created based on oscillatory expression of clock genes in presomitic mesoderm. However, the mechanism for converting the dynamic oscillation to a static pattern of somites is still unclear. Here, we provide evidence that Ripply/Tbx6 machinery is a key regulator of this conversion. Ripply1/Ripply2-mediated removal of Tbx6 protein defines somite boundary and also leads to cessation of clock gene expression in zebrafish embryos. On the other hand, activation of *ripply1/ripply2* mRNA and protein expression is periodically regulated by clock oscillation in conjunction with an Erk signaling gradient. Whereas Ripply protein decreases rapidly in embryos, Ripply-triggered Tbx6 suppression persists long enough to complete somite boundary formation. Mathematical modeling shows that a molecular network based on results of this study can reproduce dynamic-to-static conversion in somitogenesis. Furthermore, simulations with this model suggest that sustained suppression of Tbx6 caused by Ripply is crucial in this conversion.

Periodic generation of somite boundaries is regulated by a molecular oscillator, called the segmentation clock[1–4]. In zebrafish embryos, this clock is based upon oscillatory expression of members of Hairy and enhancer-of-split family genes, called *her1* and *her7*, encoding basic helix-loop-helix transcriptional repressors[5–7]. The oscillatory wave of the segmentation clock travels from the posterior to anterior presomitic mesoderm (PSM), and dynamics of the segmentation clock are thought to be converted into a stabilized pattern in the anterior PSM, resulting in periodic generation of somite boundaries[1,2]. Analyses with fixed and live zebrafish embryos revealed that the oscillation frequency slows as the oscillatory wave approaches the anterior PSM[2,8–10]. Thus, it was proposed that as the clock slows, it eventually arrests oscillating, resulting in fixation of phases of the clock. These fixed phases are thought to be reflected in the spatial pattern of somites, i.e., the rostro-caudal polarity within each somite, and the position of somite boundaries[2,8]. The idea that arrest of clock oscillation determines somite patterns has also been employed in models based on quantitative observations in other vertebrates[11,12]. However, there is little experimental evidence directly showing that arrest and phase-fixation of oscillation directly positions somite boundaries. On the other hand, live-imaging analysis with zebrafish embryos revealed that traveling of the oscillation wave stops at the anterior PSM, but the oscillation itself is not arrested, even at the anterior end where this traveling stops[13]. Therefore, careful consideration should still be given to whether oscillation arrest of the segmentation clock directs somite patterning.

On the other hand, it has been proposed that somite boundaries are defined by the anterior border of the Tbx6 protein-expressing domain[14–16], because the future somite boundary is coincident with this border. In each segmentation cycle, the Tbx6 border is newly generated by periodic degradation of Tbx6 protein via physical interaction with Ripply1 and Ripply2[16–19]. In the mouse, expression of *Ripply* genes is dependent on Mesp2[15,20], a basic-helix-loop-helix transcriptional factor. Since Notch signaling, a component of the segmentation clock in mice, activates Mesp2[21] in conjunction with Fgf/Erk signaling[14], periodical regulation of the Tbx6/Ripply system is under control of the segmentation clock. However, the molecular mechanism by which dynamics of the segmentation clock are converted into periodic stabilization of the anterior border of the Tbx6 protein domain is still unknown. In zebrafish embryos, this role of

[1]Exploratory Research Center on Life and Living Systems (ExCELLS), National Institutes of Natural Sciences, 5-1 Higashiyama, Myodaiji-cho, Okazaki, Aichi 444-8787, Japan. [2]National Institute for Basic Biology, National Institutes of Natural Sciences, 5-1 Higashiyama, Myodaiji-cho, Okazaki, Aichi 444-8787, Japan. [3]The Graduate University for Advanced Studies (SOKENDAI), 5-1 Higashiyama, Myodaiji-cho, Okazaki, Aichi 444-8787, Japan. [4]Graduate School of Natural Science and Technology, Kanazawa University, Kakuma-machi, Kanazawa 920-1192, Japan. ✉e-mail: yabe@nibb.ac.jp; uriu@staff.kanazawa-u.ac.jp; stakada@nibb.ac.jp

Mesp2 does not exist because all zebrafish *mesp* homologs are dispensable for degradation of Tbx6 protein and somite boundary definition[22]. Therefore, the clock-to-boundary conversion mechanism is even less clear.

To better understand the interaction between the segmentation clock and the Tbx6/Ripply system, we took a genetic approach using zebrafish. Surprisingly, we found that clock oscillation continues as long as Tbx6 is present, and conversely, removing Tbx6 leads to cessation of clock gene expression, i.e., the collapse of the segmentation clock. Furthermore, by precisely examining regulation of the Tbx6/Ripply system, we propose a model in which periodic activation of Ripply caused by the segmentation clock fixes the anterior border of the Tbx6 domain. Based on these results and those obtained for regulation of the Tbx6/Ripply system, we provide a mathematical model showing that a molecular network based on interaction of Ripply, Her1/Her7, Tbx6, and Fgf/Erk signaling is the minimal network capable of forming metameric patterns of somites.

## Results

### Tbx6/Ripply machinery defines somite boundaries

Two *ripply* genes, *ripply1* and *ripply2* are expressed in the anterior PSM during zebrafish somitogenesis (Fig. 1a)[16,23]. These genes serve partially redundant functions in degradation of Tbx6 proteins[16,22,24]. A *ripply1* single-mutant exhibits prolonged expression of Tbx6 protein and this expression is enhanced by loss of *ripply2* (Supplementary Fig. 1). Since the anterior border of the Tbx6 domain coincides with that of the future somite boundary, Ripply-mediated degradation seems to be involved in the definition of somite boundaries[14,16]. To confirm whether the Tbx6 border actually creates the physical somite boundary, we artificially generated Tbx6 borders by transplanting *ripply1; ripply2*

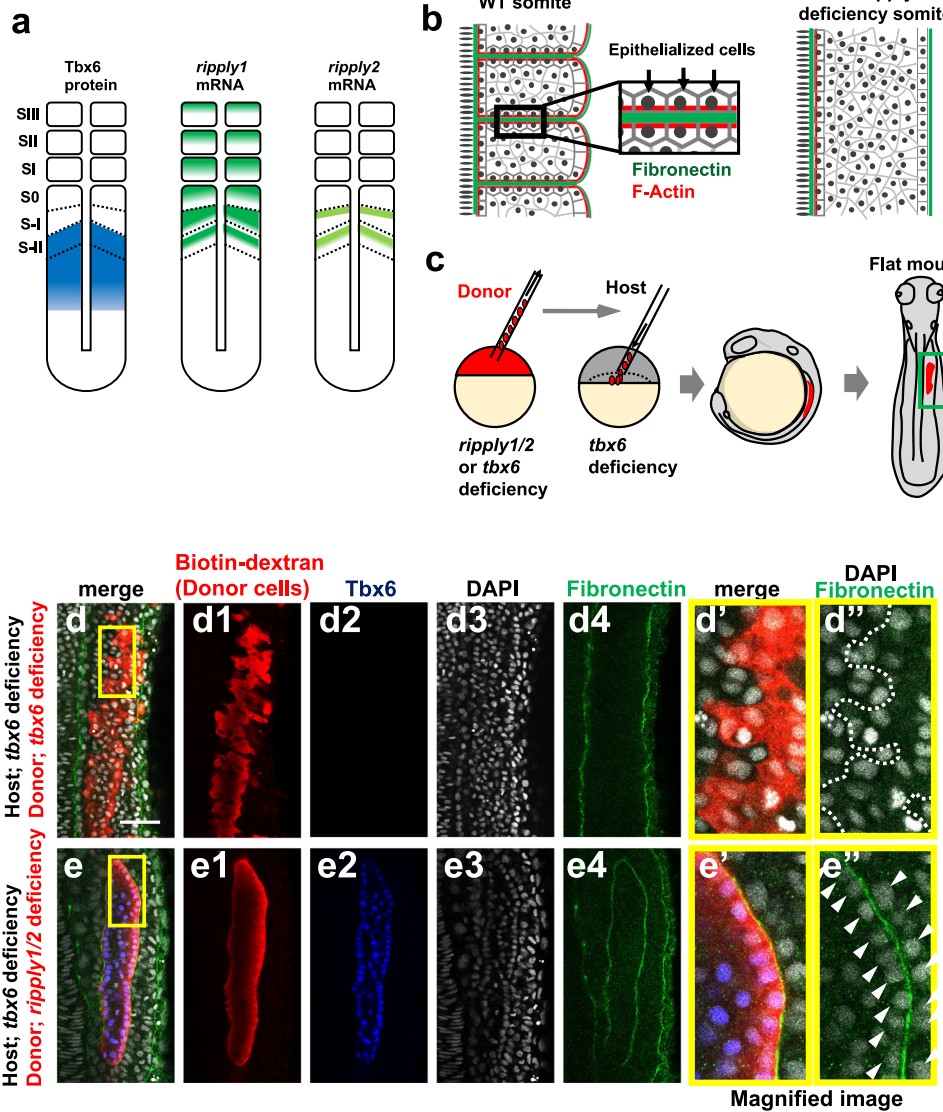

**Fig. 1 | A somite-like physical boundary is formed at the edge of the Tbx6-expressing cell mass. a** Schematic illustration of Tbx6, *ripply1* and *ripply2* expression during zebrafish somitogenesis. **b** Schematic representation of somite boundary structure in wild-type (left) and *tbx6* or *ripply1/2* double-deficient embryos (right). In wild-type embryos, fibronectin is assembled at the somite boundary. In addition, cells adjacent to somite boundaries are epithelialized with accumulation of F-Actin at the basal domain[25]. **c** Schematic representation of transplantation experiment. Rhodamine/Biotin-labeled cells from *tbx6*-or *ripply1/2* double-deficient embryos were transplanted to the lateral margin of the *tbx6* deficient host embryo at the dome-40% epiboly stage. Chimeric embryos were fixed at somite stages 8–9 and flat-mounted after IHC staining. The green box indicates the approximate area observed in (**d, e**) and Supplementary Fig. 2. **d, e** IHC staining of fibronectin (green) and Tbx6 (blue) in *tbx6*-deficient embryos with transplanted *tbx6*-deficient donor cells(**d**; *n* = 16) and *ripply1/2* double-deficient donor cells (**e**; *n* = 17). Donor cells were detected with rhodamine labeled streptavidin (red). (**d', d'', e', e''**) High-magnification images of the area indicated by yellow boxes in (**d**) and (**e**). Aligned nuclei are indicated by white arrow-heads in (**e''**). Scale bar indicates 50 μm in (**d**).

double-deficient donor cells, in which Tbx6 protein was highly stabilized[16], in *tbx6*-deficient host embryos (Fig. 1c). While no obvious boundary was formed at the edge of the donor cell mass in controls, a physical boundary resembling a somite boundary was formed around *ripply*-deficient donor cells. Similar to normal somite boundary, fibronectin accumulated and actin filaments were clearly formed along this boundary (Fig. 1b, d, e, Supplementary Fig. 2)[25,26] and the centrosomes, which are located in the apical side of epithelial cell[25], were located almost opposite this boundary (Supplementary Fig. 2). Thus, we concluded that an epithelialized somite boundary is actually formed along the Tbx6 border.

## Tbx6/Ripply machinery stops clock gene expression

In addition to boundary formation of the Tbx6 domain, *ripply* genes are involved in suppression of *her1* and *her7* expression, as previously reported in morpholino-mediated knock down experiments[23]. We confirmed this result by generating *ripply1* mutants and found that sustained expansion of *her1* and *her7* was enhanced by loss of *ripply2* (Supplementary Fig. 3). To examine whether oscillation is maintained in this sustained expression, we performed two-color FISH analysis using *her1* intron and exon probes, which detect nascent and mature transcripts, respectively (Fig. 2a–d). In wild-type embryos, each stripe of nascent *her1* transcripts was shifted more anteriorly than that of mature transcripts in the anterior PSM, showing that the oscillation wave of *her1* expression propagates from posterior to anterior (Fig. 2a, c). In *ripply1* and *ripply2* double-mutant (*ripply* dKO) embryos, this shift was maintained farther even in more anterior regions where *her1* expression was expanded. Thus, in *ripply* dKO embryos, oscillation of the segmentation clock continued far beyond the position where the future somite boundary would normally be determined (Fig. 2b, d). To examine this persistent oscillation directly by live imaging, we generated a knock-in fish in which endogenous Her7 can be visualized by tagging with three copies of the Achilles, fast-maturable YFP variant[27] (Fig. 2e–h; Supplementary Fig. 4 and Supplementary Movie 1, 2). While anterior propagation of the wave of Her7-Achilles oscillation was restricted to the PSM in wild-type embryos, this propagation continued to be observed anterior to the PSM in *ripply* dKO embryos. These results indicate that clock oscillation continues as long as Tbx6 is present. In other words, Tbx6 removal by Ripply leads to cessation of clock gene expression, i.e., the collapse of the segmentation clock.

## Cessation of clock gene expression by Ripply

Next, we examined the molecular mechanism by which Ripply collapses the segmentation clock. Fgf/Erk signaling and Wnt/β-catenin signaling, both of which forms posterior-to-anterior gradient in the posterior PSM, maintains PSM cells in an undifferentiated state[28–31]. Thus, the persistence of these signaling types in the anterior region may cause prolonged oscillation of the segmentation clock. However, in *ripply* dKO embryos, expression of *fgf8a* and *wnt8a*, as well as activation of Fgf/Erk signaling, monitored by phosphorylation of Erk and *hes6* expression[32], and activation of Wnt signaling, monitored by *axin2* and *msgn1* expression[33,34], were unchanged (Supplementary Fig. 5).

On the other hand, since Ripply degrades Tbx6 protein, it is plausible that Ripply collapses the segmentation clock by suppressing Tbx6. Previous reports suggest that Tbx6 acts as an upstream regulator of zebrafish *her1* and *her7* and mouse *Hes7* expression in the PSM[35–37]. For instance, the most anterior stripe, but not posterior stripes, of *her1* and *her7* expression are lost in zebrafish embryos defective for *tbx6*. We found that *tbx6* is epistatic to *ripply1* and *ripply2* in the regulation of *hairy*-related genes because *tbx6, ripply1*, and *ripply2* triple-homozygous mutant embryos exhibited a phenotype identical to that of *tbx6* single mutants in *her7* expression (Fig. 2i–l).

Then, we directly examined the effects of Tbx6 and Ripply on the activity of the *her1* promoter in HEK293T cells (Fig. 2m, n). We utilized a region approximately 8 kb upstream of the *her1* gene, which is sufficient to recapture endogenous dynamics of *her1* expression in the PSM[9] and which contains a Tbx6-binding sequence[38]. This promoter was activated by Tbx6, but this activation was clearly repressed by Ripply1 and Ripply2.

To test whether *her1* expression is actually associated with Tbx6 in vivo, we compared spatial dynamics of *her1* transcripts with Tbx6 proteins in the anterior PSM (Fig. 2o–t). In zebrafish embryos, Tbx6 removal does not start at the anterior border of the Tbx6 protein domain, but at a position one segment posterior to it, and then removal expands anteriorly (Fig. 2o′–q′). As a result, the most anterior part of the Tbx6 protein domain periodically appears as a thin band, and subsequently disappears[16]. Notably, we found that the wave of *her1* transcription, which propagated anteriorly during the progression of each segmentation cycle, never traveled across the anterior border of the Tbx6 protein domain or the thin anterior band. Then the most anterior stripe of *her1* transcription disappeared as Tbx6 proteins were removed (Fig. 2p, q, s, t). Thus, dynamics of *her1* transcription are closely correlated with those of Tbx6 proteins during a cycle of somite segmentation in the anterior PSM, suggesting that Tbx6 removal collapses the expression of clock genes in vivo. Taken together, we conclude that arrest of segmentation clock oscillation does not cause somite boundary formation as has been proposed[2–4,12], but rather Ripply-mediated periodical removal of Tbx6 proteins, which determines somite boundaries, causes the collapse of the segmentation clock.

## Regulation of *ripply* expression by the segmentation clock

Our results do not support previous models that the somite boundary is established by periodic fixation of the segmentation clock. If so, how is the temporal periodicity of the segmentation clock transformed into a spatial pattern of somite boundaries? To address this question, we examined the mechanism by which the expression of *ripply* genes is regulated. In mouse somitogenesis, Mesp2, a bHLH transcriptional factor, functions as an upstream activator of *Ripply1* and *Ripply2* in the anterior PSM, and this *Mesp2* expression is periodically regulated by Notch signaling, a component of the segmentation clock in the mouse[39]. In contrast, since none of zebrafish *mesp* homologs are required to activate the expression of *ripply1* and *ripply2* in the definition of somite boundaries[22], it was unclear how the expression of zebrafish *ripply1* and *ripply2* are regulated by the segmentation clock.

Thus, we first examined the initial stage of *ripply* gene expression by detecting its nascent mRNA (Fig. 3a–f). In the anterior PSM (Fig. 3a, d), but not in somites, the most posterior stripe of nascent *ripply1* transcripts exhibited a nested pattern with Her1-Venus expression. Similarly, nascent *ripply2* and *her1* transcripts and Her1-Venus protein also displayed mutually exclusive expression in the anterior PSM (Fig. 3b, c, e, f). These results suggest that Her1 negatively regulates expression of *ripply* genes in the anterior PSM.

To test this possibility, we examined *ripply1* and *ripply2* expression in *her1* and *her7* double-mutant embryos, because *her1* and *her7* are redundantly required in zebrafish somitogenesis[6,7] (Fig. 3g–j). We generated new alleles of *her1* and *her7* mutations by TALEN-mediated mutagenesis, and embryos homozygous for these *her1* and *her7* alleles (*her* dKO) displayed severe defects in somite boundary formation, as previously reported with embryos in which a chromosome region covering *her1* and *her7* loci was deleted[7] (Supplementary Fig. 6). While *ripply1* and *ripply2* were expressed in a two-striped, or sometimes a one-striped manner in the anterior PSM of normal embryos, repression of their expression in the inter-stripe region was prevented in *her* dKO embryos. In addition, overexpression of *her1* under control of the *hsp70* promoter induced severe reduction of *ripply1* and *ripply2* expression in the anterior PSM (Fig. 3k–n). Thus, *her1* and *her7* negatively regulate *ripply1* and *ripply2* expression in the anterior PSM during somitogenesis.

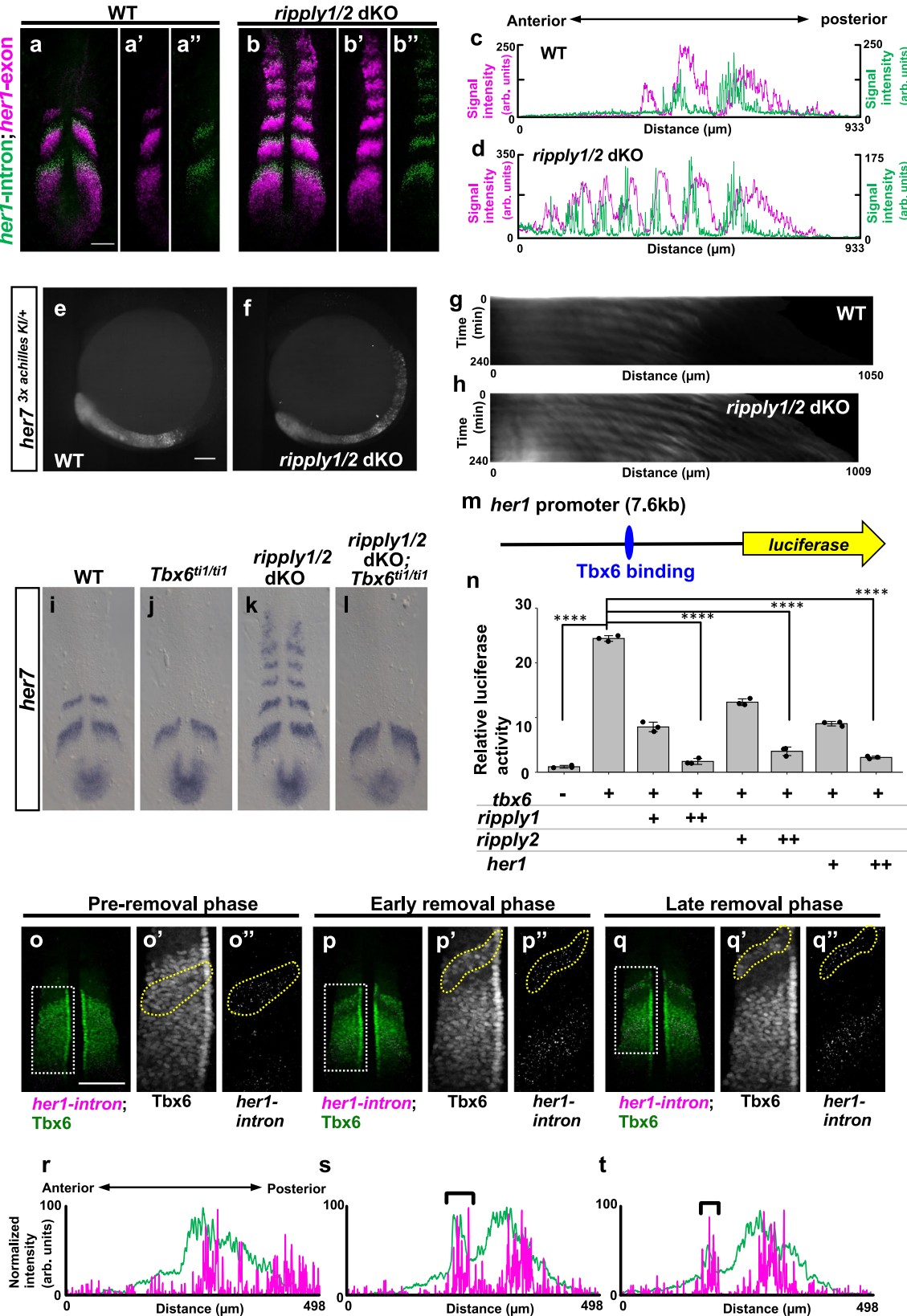

Furthermore, we examined effects of Her1 on transcriptional activity of the *ripply2* promoter in HEK293T cells (Fig. 3o, p). We isolated a region 8 kb upstream of the *ripply2* gene, which was sufficient to recapture endogenous *ripply2* expression in embryos (Supplementary Fig. 7). This promoter was activated by Tbx6, probably through a Tbx6-binding sequence in the proximal region. This Tbx6-dependent activation is also supported by previous CHIP-seq analysis (NCBI accession number; GSE57332)[40] and by the fact that significant reduction of *ripply1* and *ripply2* transcription was observed in *tbx6/fss* mutant embryos[23] (Supplementary Fig. 8). Notably, Her1 repressed Tbx6-dependent activation of the *ripply2* promoter, depending on putative *her1* binding sites[41], suggesting that Her1 represses *ripply*

**Fig. 2 | Regulation of segmentation clock arrest by Ripply-mediated removal of Tbx6 protein. a, b** Double in situ hybridization analysis of mature *her1* mRNA (magenta) and nascent *her1* mRNA (green), detected with exon and intron probes, respectively, at the 6-somite stage of WT (**a**, *n* = 3) and *ripply1^{kt1032/kt1032}*; *ripply2^{kt1034/kt1034}* embryos (**b**, *n* = 3). Scale bar indicates 100 μm. **c, d** Signal intensity plots of **a** and **b** with anterior toward left. **e–h** Fluorescent signal of Achilles (**e**; *n* = 2, **f**; *n* = 3) and kymograph of dynamics of Her7-3xAchilles oscillation in WT (**g**) and *ripply1^{kt1032/kt1032}*; *ripply2^{kt1034/kt1034}* (**h**) embryo carrying *her7^{3xachilles-KI}* heterozygous alleles. Scale bar indicates 100 μm. The posterior end of the PSM is aligned at the left side at all time points. **i–l** Epistatic analysis of *ripply1/2* and *tbx6* in regulation of segmentation clock gene expression at the anterior paraxial mesoderm. *her7* mRNA expression in WT (**i**; *n* = 6), *tbx6^{tl/tl}* (**j**; *n* = 7), *ripply1^{kt1032/kt1032}*; *ripply2^{kt1034/kt1034}* (**k**; *n* = 3) and *ripply1^{kt1032/kt1032}*; *ripply2^{kt1034/kt1034}*; *tbx6^{tl/tl}* (**l**; *n* = 2) at the 6-somite-stage. **m, n** Reporter assay showing the effects of Tbx6 and Ripply on activity of *her1*

promoter in HEK 293T cells. A schematic illustration of a luciferase reporter construct is shown in **m**. Crosses under each bar indicate amounts of plasmids used for transfection. The average of normalized firefly luciferase activity with pCS2 + (mock) alone was set at 1. Error bars represent standard deviations (*n* = 3 in each experiment). Differences in relative luciferase activity were statistically evaluated using one-way ANOVA followed by the Tukey–Kramer test. ****$p < 0.0001$. **o–t** Double staining of Tbx6 protein (green) and *her1* nascent mRNA (magenta) at the 6–7 somite-stage. The phase of somitogenesis was estimated by the degree of Tbx6 protein removal in the future somite area (**o**; *n* = 5, **p**; *n* = 3, **q**; *n* = 4). Magnified images of individual channels surrounded by white dotted squares are shown in the right. Areas surrounded by yellow dotted lines indicate the most anterior stripe of *her1* transcription. Scale bar indicates 100 μm. Relative intensity of Tbx6 and *her1* shown in (**o**, **p**, **q**) is plotted in (**r**, **s**, **t**), respectively, with anterior toward left. Black brackets indicate the position of the anterior band of Tbx6 protein.

transcription by direct binding to the *ripply2* promoter. These in vitro and in vivo data support the idea that the two clock genes, *her1* and *her7*, negatively regulate expression of *ripply* genes.

## Regulation of *ripply* expression by Fgf/Erk signaling

In addition to the segmentation clock, signaling in the posterior PSM is involved in somite segmentation[1,29,30]. Although our previous study showed that inhibition of Fgf signaling causes a posterior shift of the *ripply1*-expressing domain in zebrafish embryos[16], details of their interaction remained unclear. Thus, we also examined the effect of Fgf/Erk inhibition on *ripply* gene expression in zebrafish embryos. Since an effect of Fgf/Erk signaling on the segmentation clock has been implied[42,43], we avoided this effect by using *her* dKO embryos (Fig. 4a–h). Although the striped expression pattern of *ripply* genes was impaired in the anterior PSM, *ripply1* and *ripply2* expression was still evident in *her* dKO embryos. In control embryos, nascent transcripts of *ripply1* and *ripply2* were detected only in the anterior half of the Tbx6 protein domain (Fig. 4a, c, e, g). In contrast, after 15 min of treatment with MEK1 inhibitor, which was sufficient to abolish detectable Erk signaling throughout the PSM, *ripply1-* and *ripply2*-expressing domains expanded to the posterior end of the Tbx6 protein domain (Fig. 4b, d, f, h). Thus, Erk signaling restricts the *ripply*-expressing domain to the anterior half of the Tbx6 protein domain in a manner independent of *her1* and *her7*. In addition, in *msgn1: ggff/uas-ca-xmek1* transgenic embryos, in which Erk signaling was upregulated throughout the paraxial mesoderm, *ripply1* and *ripply2* expression was severely reduced, despite increased expression of *tbx6* (Fig. 4i–r). These results indicate that Erk signaling represses *ripply* expression downstream of or in parallel with function of Tbx6. Consistent with this, *tbx6*-dependent activation of the *ripply2* promoter was repressed by activation of Erk signaling in HEK293T cells (Fig. 4s).

## Dynamics of *ripply* expression and future somite boundaries

Since our finding implies that Hairy-related transcriptional repressors downregulate expression of *ripply1* and *ripply2* in conjunction with Fgf/Erk signaling, we examined dynamics of *ripply1* and *ripply2* expression, comparing them with that of Her1 and phosphorylation of Erk by combined analysis using FISH and multi-color IHC (Fig. 5). As previously described, expression of *ripply1* and *ripply2* is periodically activated in the anterior PSM. Just prior to onset of *ripply* gene expression, the Her1 protein domain was located in the anterior region of the active Erk (dually phosphorylated Erk; dpErk) gradient (Fig. 5a, g). Subsequently, with anterior progression of the Her1 oscillatory domain, *ripply1* and *ripply2* transcription was initiated at the anterior side of the Erk activity gradient (Fig. 5b, h). Then, the *ripply* expression domain moved anteriorly, correlated with anterior propagation of the Her1 oscillation wave (Fig. 5c, a, i, g). During this phase, the next wave of Her1 pushes the posterior edge of the *ripply* expressing domain anteriorly. Thus, *ripply* expression is initially activated in the gap

between Her1 waves in the anterior skirts of the Erk activity gradient, and this activation occurs periodically during anterior propagation of the Her1 wave (Fig. 5m). Since the amplitude of Her1 oscillation increases as the wave moves into the anterior PSM[9,10,13], this elevation probably serves to strongly suppress *ripply* expression in the anterior PSM, where Erk-mediated suppression of *ripply* expression does not occur.

To examine how periodic activation of *ripply* expression fixes the anterior border of the Tbx6 protein domain, we next compared spatial dynamics of Ripply proteins with Tbx6 proteins in the anterior PSM. To detect endogenous Ripply1 and Ripply2 protein, knock-in fishes, in which a V5-tag was added to Ripply1 or Ripply2 in the N-terminal region, less conserved among species, was generated with CRISPR/Cas9 (Supplementary Fig. 9). Genetic analysis confirmed that addition of a V5-tag did not impair Ripply activity. This visualization of Ripply proteins showed that, as predicted, Tbx6 removal was initiated in correlation with the expression of Ripply1 and Ripply2 proteins (Supplementary Fig. 10). Importantly, after the onset of Ripply1 and Ripply2 protein expression (Supplementary Fig. 10b, d), the posterior edge of the Ripply protein domain gradually moved away from the newly formed anterior edge of the Tbx6 protein domain (Fig. 5n, o; Supplementary Fig. 10c, e, f). This finding suggests that Ripply expression induces Tbx6 removal in the anterior PSM, but once removed, Tbx6 suppression is stably maintained even after the reduction of Ripply proteins.

Then, we examined whether transient Ripply expression triggers persistent suppression of Tbx6 by utilizing a mosaic transgenic approach. As shown in Fig. 6a, HA-tagged Ripply1 whose expression is inducible by the *hsp70* promoter was sparsely expressed in a *ripply*-defective background (Fig. 6a, b). In these transgenic cells, which can be traced by the remaining expression of GFP, transiently expressed HA-tagged Ripply1 protein was not stable, like endogenous Ripply proteins, and was cleared within the 90-min after heat shock treatment, but repression of Tbx6 persisted more than 150 min (Fig. 6c–h). Given that the transition from creation of a new Tbx6 boundary at S-II to the formation of a morphological boundary at S0 takes 60 min, the period of suppression of Tbx6 expression shown in this experiment is sufficient to maintain the boundary until the formation of a morphological boundary.

Recently, it was shown that expression of *tbx6* is directly activated by Tbx6 itself in the anterior PSM of wild-type zebrafish embryos and the anteriorly expanded Tbx6 expressing domain of *ripply* dKO embryo, and that this positive feedback loop (PFL) is essential for *tbx6* expression after its initiation[44]. Given that a PFL is a bistable switch to generate distinct cellular states with high and low expression, once Ripply triggers a decrease in Tbx6 protein, the PFL of Tbx6 finally stabilizes this expression at a low level. We examined this point in detail in the mathematical modeling analysis described below. Finally, we note that *ripply1* expression was further increased even after a new anterior edge of the Tbx6 protein domain was created (Supplementary Fig. 10A–C).

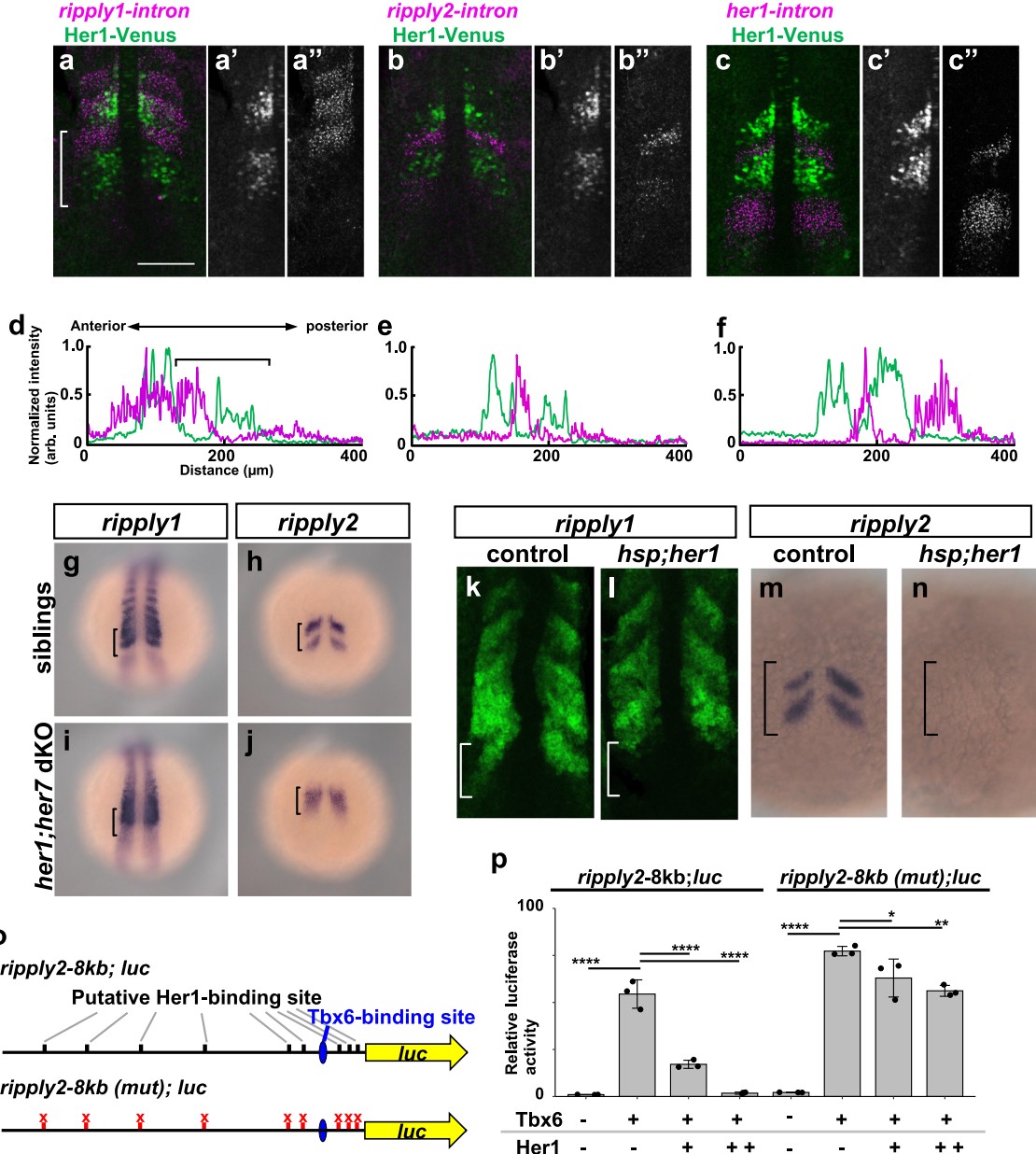

**Fig. 3 | Regulation of *ripply1/2* expression by the Hairy-related transcriptional repressor. a–f** Comparison of *ripply1*, *ripply2* and *her1* transcribing areas with Her1 protein-expressing area. The Her1 protein-expressing area was represented by IHC using anti-GFP antibody in *TG(her1:her1-venus)* heterozygous embryos (green). *ripply1*, *ripply2* and *her1*-expressing areas are represented by FISH using intron probes for *ripply1* (**a**; *n* = 8), *ripply2* (**b**; *n* = 8) and *her1* (**c**; *n* = 4) to detect nascent mRNAs, respectively (magenta). Left-half images of individual channels are shown in right panels. A white bracket indicates the position of the anterior PSM. Scale bar indicates 100 μm. **d–f** Signal intensity plots of (**a–c**) with anterior toward the left. A black bracket indicates the position of the anterior PSM. **g–j** Analysis of *ripply1* (**g**, **i**) and *ripply2* (**h**, **j**) expression in *her1;her7* double-mutant embryo. WT (**g**; *n* = 15, **h**; *n* = 16) and *her1^{k1060/kt1060}; her7^{kt1061/kt1061}* (**i**; *n* = 10, **j**; *n* = 11) embryos were fixed at 8–9 somite stage. **k–n** Analysis of effects of *her1* overexpression on the *ripply1*

(**k**; *n* = 7, **l**; *n* = 5) and *ripply2* (**m**; *n* = 11, **n**; *n* = 13) expression. Embryos obtained by crossing *Tg(hsp:her1)/+* males with wild-type females were fixed at the 8–9-somite stage after 30 min incubation at 37 °C. **o, p** Reporter assay showing effects of *her1* expression on the activity of the *ripply2* promoter using HEK293T cells. **o** Schematic image of a reporter construct used for the reporter assay. The putative Her1/Her7 binding site indicates previously identified zebrafish Her1/Her7 consensus binding sequences, CACGNG. In the *ripply2-8kb* (mut)-luc construct, all CACGNG motifs were substituted to AAAGNG or AAAAAA. Crosses under each bar indicates amounts of plasmids used for transfection. Average of normalized firefly luciferase activity with pCS2+ (mock) alone was set at 1. Error bars represent standard deviations (*n* = 3 in each experiment). Differences of relative luciferase activity were statistically evaluated using one-way ANOVA followed by the Tukey–Kramer test. **p* = 0.03, ***p* = 0.001 *****p* < 0.0001.

This increase in *ripply1* expression may help to maintain the newly established edge of the Tbx6 protein domain.

## Model of an essential network in somite boundary definition

In this study, we identified three interactions; (1) Repression of *her1/her7* expression through removal of Tbx6 protein by Ripply, (2) Repression of *ripply* expression by direct binding of Hairy-related

transcription factors to their promoter, (3) Repression of *ripply* expression by Erk signaling. Combining these newly identified interactions with already known gene regulation mechanisms[8,40,44], we propose a gene network model for regulation of somite boundary formation and clock arrest (Fig. 7a).

To ascertain the validity of this network model, we converted it into a mathematical model (Supplementary Fig. 11). In this

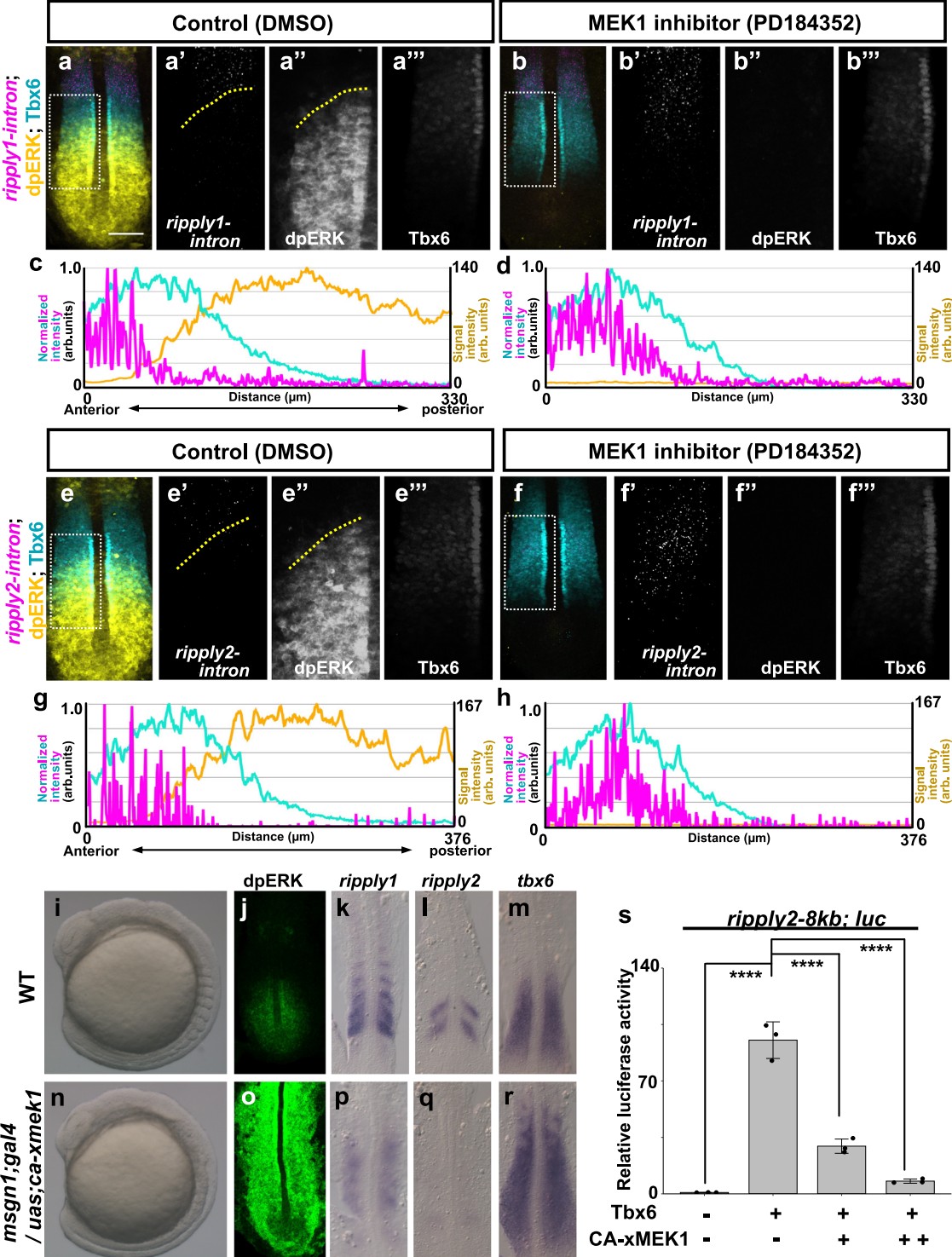

**Fig. 4 | Regulation of *ripply1/2* expression by FGF/ERK signaling. a–h** Effect of ERK inhibition on expression of *ripply1* (**a**–**d**) and *ripply2* (**e**–**h**). Embryos were fixed at the 8-somite stage after 15 min treatment with 0.01% DMSO (**a**; *n* = 4, **e**; *n* = 5) or 10 μM PD184352 (**b**; *n* = 5, **f**; *n* = 5). The area of *ripply* transcription was represented by FISH using an intron probe (magenta) with co-staining of Tbx6 protein (cyan) and dpERK (yellow) by IHC. Magnified images of individual channels surrounded by white dotted squares are shown in the right panel. Yellow dotted lines indicate the anterior end of the dpERK signal gradient in the control embryo. Quantified signal intensity of (**a**, **b**, **e**, **f**) was plotted in (**c**, **d**, **g**, **h**) with anterior toward the left. Scale bar indicates 100 μm. **i**–**r** Effects of ERK over-activation on somitogenesis and *ripply1* and *ripply2* expression. Live phenotype of control WT (**i**) and *msgn1;gal4/*

*uas;ca-xmek1* (**n**)embryo at the 8-somite stage. WT (**j**–**m**) and *msgn1;gal4/uas;ca-xmek1* (**o**–**r**) embryos were fixed at the 8-somite stage and stained dpERK (**j**; *n* = 3, **o**; *n* = 3) with IHC or *ripply1* (**k**; *n* = 8, **p**; *n* = 8), *ripply2* (**l**; *n* = 10, **q**; *n* = 8) and *tbx6* (**m**; *n* = 8, **r**; *n* = 7) mRNA by WISH. Embryos were mounted after yolk removal with heads toward the upper side. **s** Reporter assay showing effects of ERK over-activation on activity of the *ripply2* promoter using HEK293T cells. Crosses under each bar indicate amounts of plasmids used for transfection. The average of normalized firefly luciferase activity with pCS2+ (mock) alone was set at 1. Error bars represent standard deviations (*n* = 3 in each experiment). Differences of relative luciferase activity were statistically evaluated using one-way ANOVA followed by the Tukey–Kramer test. ****$p < 0.0001$.

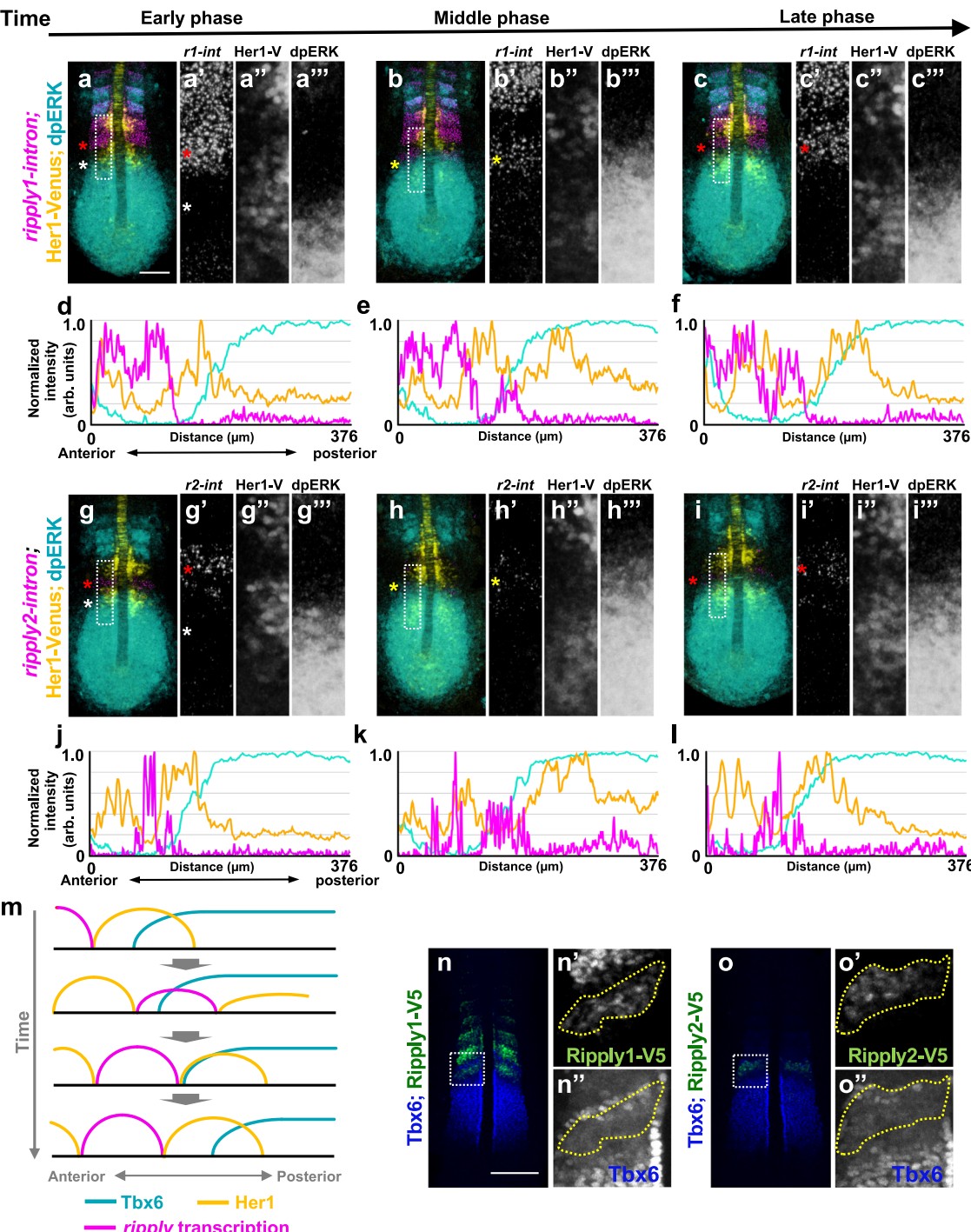

**Fig. 5 | Dynamics of expression of proteins and mRNA involved in definition of the somite boundary during somitogenesis. a–l** series of fixed *tg(her1:her1-venus)* heterozygous embryos at the 7-somite stage stained Her1-venus protein (yellow) and dpERK (cyan) with *ripply1* (magenta; **a–c**) and *ripply2* (magenta; **g–i**) nascent mRNA. Magnified images of individual channels surrounded by white dotted squares are shown in the right panel. Phase was estimated by the relative position of Her1-venus stripes and anterior end of the dpERK signal gradient (**a**; *n* = 10, **b**; *n* = 8, **c**; *n* = 12, **g**; *n* = 9, **h**; *n* = 7, **i**; *n* = 9). Quantified signal intensity of (**a–c**, **g–i**) was plotted in (**d–f**, **j–l**) with anterior toward the left. **m** Schematic illustration of dynamics of Her1, Tbx6 proteins, and *ripply1* and *ripply2* transcription during zebrafish somitogenesis. **n**, **o** Double staining of Tbx6 protein (Blue) with Ripply1-V5 (**n**; green) or Ripply2-V5 (**o**; green). *ripply1^{u5/+}* and *ripply2^{u5/u5}* embryos were fixed at the 6–7 somite stage (**n**; *n* = 8, **o**; *n* = 9). Magnified images of individual channels surrounded by white dotted squares are shown right. Areas surrounded by a yellow dotted line indicate the most posterior domain of Ripply protein expression. Scale bar indicates 100 μm.

mathematical model, the PSM is described as a one-dimensional array of cells (Supplementary Fig. 11a, b). Its left end is where somites differentiate and its right end is at the embryonic tailbud. To model posterior progression of the tailbud, positions of the left and right ends of the array move rightward at a constant speed. Based on previous work[45–47], the time evolution of mRNA and protein concentrations is described with delay differential equations (Supplementary Fig. 11c). The model distinguishes nascent and mature mRNAs of *her* and *ripply* as separate variables to compare simulations with experimental results. The model also includes concentrations of Her, Ripply, Tbx6, and dpErk proteins, incorporating transcriptional and post-translational regulation between these transcription factors

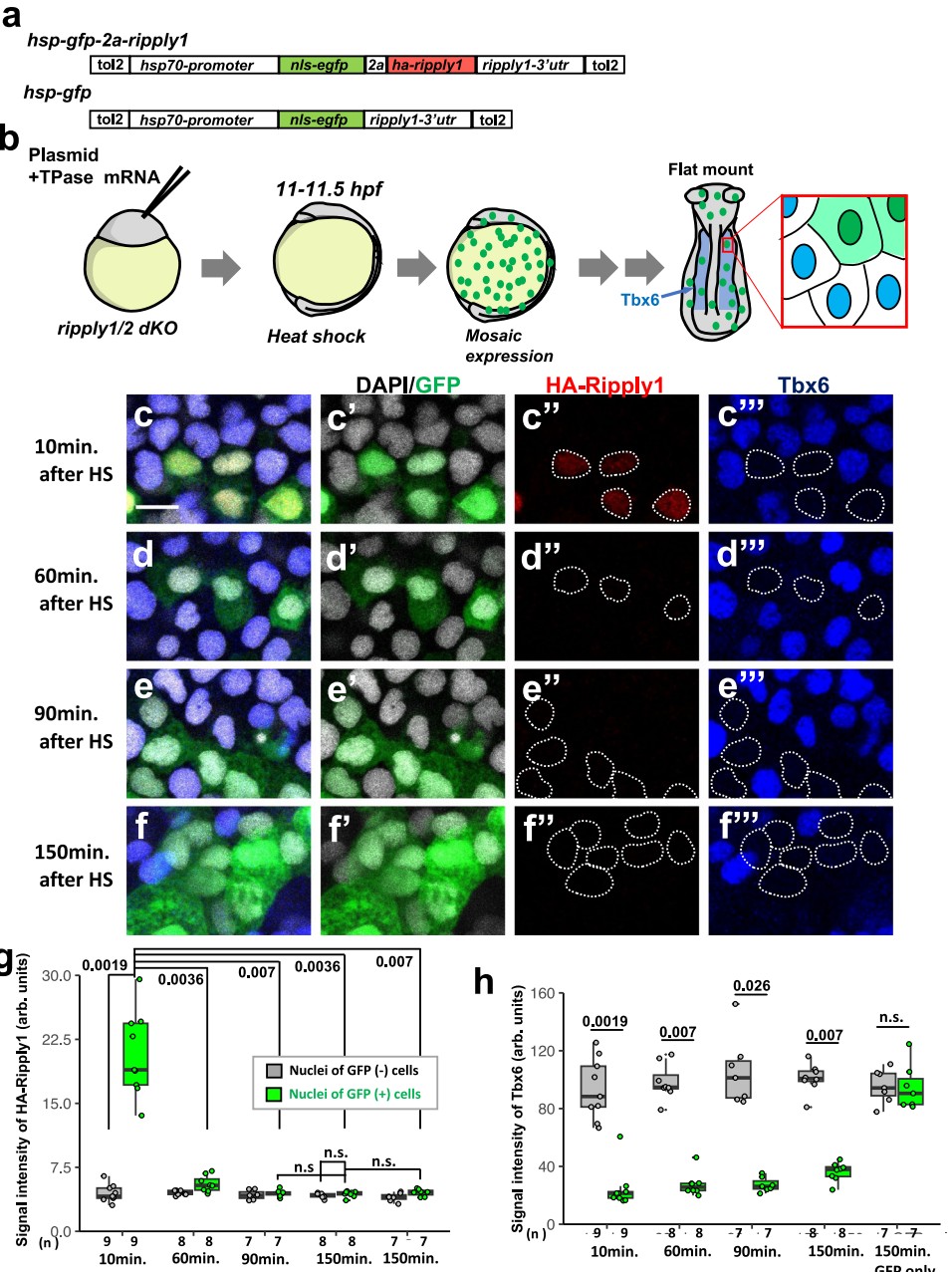

**Fig. 6 | Repression of Tbx6 is sustained for almost two somite cycles by transient expression of Ripply in *ripply* dKO embryos. a** Schematic representation of DNA constructs. nls-EGF and HA-tagged Ripply were separately expressed by 2A-peptide under-control of the *hsp70* promoter. A Tol2 inverted repeat sequence was introduced on both sides of the insertion to improve efficiency of genomic integration. **b** Schematic representation of the transient *ripply* expression assay in the anterior paraxial mesodermal region of *ripply* dKO embryo. Twenty-five pg of plasmid DNA were injected into the *ripply1^{ktl032}* and *ripply2^{ktl034}* double-homozygous eggs. Embryos were incubated in a 38 °C water bath for 30 min. at 11 hpf (3-somite stage) to achieve mosaic expression of transgenes. Expression of Tbx6 and HA-Ripply1 were assessed in anterior paraxial mesoderm, in which ectopic expression of Tbx6 is maintained at *ripply* dKO embryos, after 10–150 min incubation at 28 °C. **c–f** IHC staining of GFP (green), HA-Ripply1 (red) and Tbx6

(blue) in the anterior paraxial mesodermal region of *ripply* dKO embryo at 10, 60, 90, and 150 min. after HS treatment (**c–f**). Nuclei were counterstained with DAPI. Individual channels are shown at right. Areas surrounded with dotted white lines indicate the position of nuclei of GFP-expressing cells. Scale bar indicates 10μm. **g, h** A statistical summary of this experiment. The intensity of HA-Ripply1 (**g**) and Tbx6 (**h**) was quantified in nuclei indicated by DAPI staining. The group of GFP only indicates the embryos injected *hsp-gfp* construct represented in (**a**) as a negative control. Box plots of medians of signal intensity in each embryo show the first and third quartile, the median is represented by a line, whiskers indicate the minimum and maximum, and outliers are shown as dots outside the box. Differences of signal intensity were evaluated by pairwise comparisons using the two-sided Wilcoxon rank sum test adjusted for multiple comparison with Bonferroni's method. *p-values are shown at the top of boxes.

(Fig. 7a). To reproduce the spatial pattern of Tbx6 protein expression, which is restricted to the middle of the PSM, we assumed an additional regulatory network that activates Tbx6 synthesis if the intensity of posterior Fgf/Erk signaling is moderate (incoherent feedforward loop[48,49]; Supplementary Fig. 11d). In addition, based on experimental

evidence[44], we introduced a PFL of Tbx6 to sustain its expression in the anterior PSM. In the posterior PSM, transcription of *her* mRNA occurs even in the absence of Tbx6 protein. This is because other Tbox transcription factors, such as *tbxta* and *tbx16*, may induce transcription of *her* mRNA[50]. Therefore, we assumed that *her* transcription requires

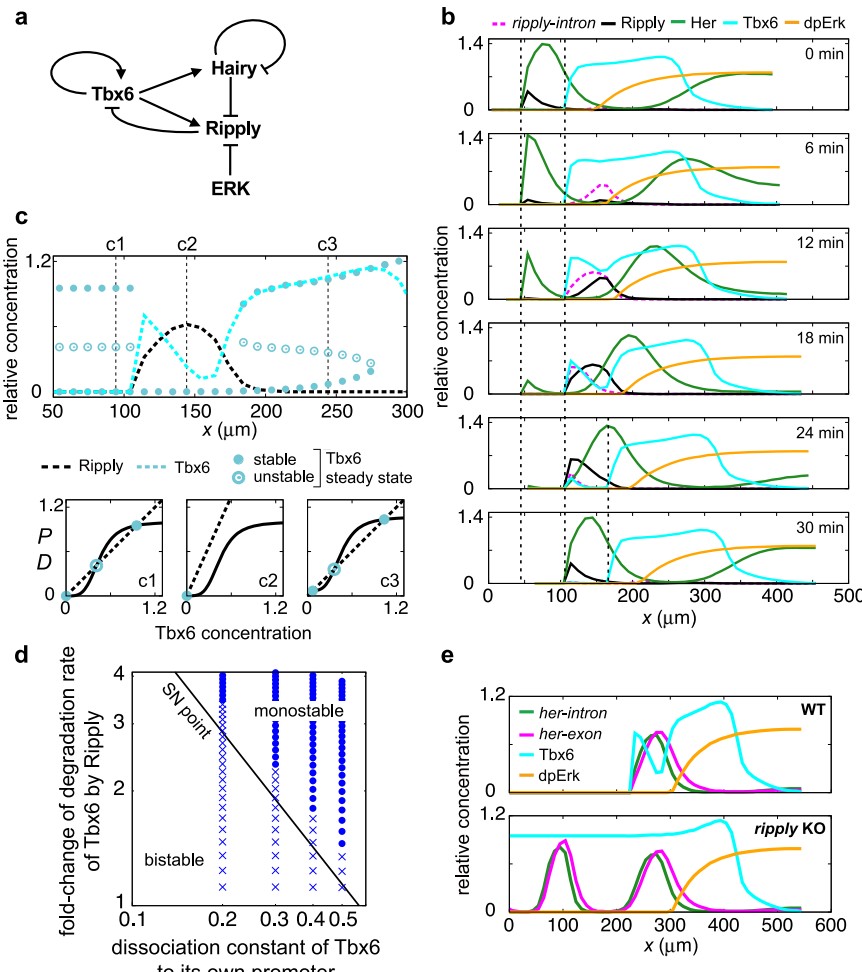

**Fig. 7 | Mathematical model showing dynamic-to-static conversion in zebrafish somitogenesis. a** Schematic diagram of the gene regulatory network in the dynamic-to-static conversion in zebrafish somitogenesis. **b** Snapshots of *ripply-intron* mRNA, and those of Ripply, Her, Tbx6, and dpErk proteins in wild-type simulation. Dotted vertical lines indicate anterior Tbx6 boundaries. Anterior is on the left. Details of changes in the expression of each factor are provided in the text. **c** Tbx6 steady states as a function of spatial position. Cyan and black broken lines indicate a snapshot of Tbx6 and Ripply protein levels, respectively, obtained by simulation ($t = 18$ min in **b**). **c1–c3** Dependence of the production speed $P$ of Tbx6 (solid line) and its degradation speed $D$ (broken line) on Tbx6 levels at each spatial position indicated by numbers (1–3) in the top panel. Circles indicate the intersection of these two lines. Filled and open circles indicate stable and unstable Tbx6 steady states, respectively. Bifurcations across the entire PSM region are

shown in Supplementary Fig. 14. **d** Phase diagram of Tbx6 boundary formation. The horizontal axis is the dissociation constant of Tbx6 protein for its own promoter ($K_6$ in Eq. (5a) in Methods), representing the strength of PFL. The vertical axis is the maximum change of the Tbx6 degradation rate by Ripply ($1 + \eta p_r$ in Eq. (5a) in Methods). To change the Tbx6 degradation rate, the Ripply translation rate ($v_3$ in Eq. (4) in Methods) was varied. Both axes are in log scale. Circles indicate correct Tbx6 boundary formation. Crosses indicate failure in Tbx6 boundary formation until a wave of Ripply travels distance of one somite (6 cells). The black line indicates saddle-node bifurcation point (SN point) at which the number of Tbx6 steady states changes. (**e**) Snapshots of spatial patterns of *her* mRNAs, Tbx6 and dpErk proteins for (top) wild-type and (bottom) *ripply* knock-out simulations. Parameter values used in simulations are listed in Supplementary Table 1.

Tbx6 protein only in the anterior PSM. To generate gene expression waves of *her*, a spatial gradient of translation delays of Her protein is imposed, based on a previous experiment[46]. Increasing translation delay lengthens peak-to-peak intervals of Her oscillation in the anterior PSM, as observed in embryos[13] (Supplementary Fig 12). For simplicity, we imposed a spatial dpErk gradient that continuously moves with the tailbud. Details of the mathematical model are described in the Methods.

Numerical simulations indicate that the proposed gene regulatory network (Fig. 7a) captures the periodic generation of anterior borders of the Tbx6 protein domain and stop of clock wave traveling. Expression of *ripply* mRNA starts at the anterior edge of the dpErk gradient, and then moves anteriorly along with arrival of a Her expression wave from the posterior PSM (Fig. 7b; 0–18 min; Supplementary Movie 3). Subsequent expression of Ripply protein traces the spatio-temporal pattern of its mRNA (Fig. 7b; 12–24 min).

Consequently, degradation of Tbx6 protein starts near the edge of the dpErk gradient and proceeds anteriorly, leaving a new border of Tbx6 expression behind (Fig. 7b; 18 min). Thus, the time difference of Tbx6 removal between cells near the dpErk edge and those near the previous Tbx6 anterior border increases with slowing of Her expression waves (Supplementary Fig. 13). In simulations, future somite regions can be defined as a spatial interval between the previous (one cycle before) and present anterior borders of the Tbx6 protein domain (dotted lines in Fig. 7b). A wave of Her protein expression reaches the previous anterior border of the Tbx6 domain, and forms a spatial gradient in a future somite region (Fig. 7b; 0–18 min), being consistent with experimental observations by Shih et al. 2015[13]. Finally, *her* expression becomes almost zero as Tbx6 protein is removed by Ripply protein.

Dynamics of Tbx6 expression across the PSM can be understood in terms of bifurcations of its steady states (Fig. 7c; Supplementary Figs. 14,

15). The PFL of Tbx6 can generate two stable and one unstable steady states as its S-shaped production and linear degradation terms intersect at three points (Fig. 7c3). Cells at the posterior end of the PSM exhibit low Tbx6 expression. As cells leave the posterior end, the Fgf/Erk gradient activates Tbx6 expression and its levels elevate to a high steady-state value through the PFL (Fig. 7c1–3; Supplementary Fig. 14). Before a pulse of Ripply protein expression, three steady states are present in the anterior PSM (Supplementary Figs. 15a). A pulse of Ripply protein expression increases the degradation rate of Tbx6, resulting in disappearance of the higher stable steady state (Fig. 7c2; Supplementary Figs. 15a, b). This disappearance occurs through a saddle-node bifurcation where the higher stable steady state and unstable one collide and disappear[51]. A wave of Ripply expression causes a sequence of the saddle-node bifurcations across cells (Supplementary Fig. 15b). To degrade Tbx6 to the low steady state, the change of the Tbx6 degradation rate by Ripply needs to be larger than the value required for a saddle-node bifurcation (Fig. 7d, Supplementary Figs. 15c, 16). Remarkably, abrupt transition of Tbx6 levels along with loss of the high steady state results in a sharp Tbx6 expression boundary. Once Tbx6 protein levels become low, cells stay at the low steady state even after removal of Ripply protein (Fig. 7c1; Supplementary Fig. 15d), consistent with Ripply expression experiments under a *ripply* dKO background (Fig. 6). The decrease in Tbx6 levels leads to the cessation of *her* expression, arresting its oscillation at the anterior PSM in the mathematical model (Supplementary Fig. 17). If the PFL is absent, Tbx6 levels recover after removal of Ripply protein, failing in boundary formation (Supplementary Fig. 18; Supplementary Movie 4). Thus, the mathematical model reveals bifurcations that Tbx6 levels undergo across the PSM.

Next, we examined whether the mathematical model could also reproduce anomalies in embryos in which components in Fig. 7a are perturbed. To simulate *ripply* dKO embryos, the transcription rate of *ripply* mRNA was set to zero (Fig. 7e, Supplementary Movie 5). In the absence of *ripply*, Tbx6 protein remains at a high steady state value in the anterior region due to the PFL. As a result, expression waves of *her* mRNA persist as in *ripply* dKO mutants (Fig. 2b, f, h), while this wave stops at the anterior border of the Tbx6 protein domain, and *her* expression disappears as Tbx6 degrades in control (Fig. 2a, e, g, p, q). We also simulated *her* dKO mutants by setting the transcription rate of *her* mRNA to zero (Supplementary Fig. 19 and Supplementary Movie 6). In this case, *ripply* expression continuously occurs at the anterior border of the dpErk gradient. In addition, the reduction of Erk phosphorylation, which models the effect of a MEK inhibitor, extends the expression domain of *ripply* posteriorly. Thus, the mathematical model reproduces effects of perturbation of the regulatory network.

It was reported that the anterior boundary of dpErk suddenly shifts posteriorly near one segment length within one cycle of segment formation in zebrafish[52–54]. Even in the presence of this shift, the proposed gene regulatory network can periodically determine somite boundaries and reproduces spatio-temporal expression patterns of components observed in embryos (Supplementary Fig. 20 and Supplementary Movie 7). Taken together, the proposed gene regulatory network is sufficient for repetitive generation of the anterior border of the Tbx6 expression domain and simultaneous termination of the clock. Through this network, the segmentation clock and Erk activity gradient cooperatively switch Tbx6 bistable states to determine future somite boundaries.

## Discussion

Since Cooke and Zeeman proposed the "clock and wavefront" model in 1976, numerous models have been proposed to explain the mechanism by which the temporal periodicity of the segmentation clock is converted to the metameric pattern of somites[1]. Although it is intuitive to define somite boundaries at stabilized positions by transition of the segmentation clock from an oscillatory to a fixed state, a live-imaging analysis with zebrafish embryos[13] and mathematical

simulation based on the regulatory network identified in this study clearly show that the segmentation clock controls somite boundaries without fixation of expression waves in space. Rather, as a result of the position of the somite boundary determined by Tbx6 degradation, the segmentation clock collapses (Supplementary Fig. 21). Our mathematical simulation shows that the molecular circuit identified in this study is the core network sufficient to reproduce the conversion of dynamics of the segmentation clock to the static pattern of somites.

One of the highlights of this study is that this core network can create bistable states of Tbx6 expression. Involvement of bistability and bifurcation of cell steady states in somite formation have been discussed previously[3,55–57]. Theoretical studies showed that the bistability of cell differentiation states could arise by mutual repression of opposing spatial gradients of retinoic acid and Fgf signaling[58] and/or by a PFL of dpErk[54]. These theoretical models further hypothesized that the segmentation clock perturbs cell states to transit from the undifferentiated state to the differentiated state by, for example, affecting the phosphorylation rate of Erk[54]. However, these studies leave some ambiguities in the molecular process by which dynamics of the segmentation clock are converted into the pattern of somites. In contrast, our mathematical model is based on a molecular network supported by experimental data showing the function and spatio-temporal expression order of key components in this conversion. In this model, Her oscillation perturbs Tbx6 steady states by allowing transient induction of Ripply expression. Subsequently, the Ripply-mediated decrease in Tbx6 causes the PFL to stop producing Tbx6, resulting in low Tbx6 levels even after Ripply expression ceases. Removal of Tbx6 protein by Ripply stops Her1 expression as previously reported with a Her1 live reporter[13]. Thus, our study suggests that bistability of Tbx6 underlies the dynamic-to-static conversion, including somite boundary formation and termination of the segmentation clock in zebrafish somitogenesis. The segmentation clock and the Erk activity gradient cooperatively regulate Ripply expression, which can switch the bistable states of Tbx6.

In this study, we show that the segmentation clock is collapsed by Ripply-induced removal of Tbx6. Because *ripply* expression is regulated by Erk signaling, which exhibits graded activation along the A-P axis, spatial information created by Erk gradient appears to control cessation of the segmentation clock and boundary formation through activation of *ripply* genes. In contrast, a recent study proposed that the cell-intrinsic timer determines the timing of oscillation arrest of the segmentation clock using dissociated cell culture system[59]. This controversy can be explained by considering the specific nature of Fgf8/Erk gradient in the PSM. In chick embryos, Fgf8 transcription occurs only at the posterior end of the PSM and the Fgf8/Erk gradient is mainly generated by gradual decay of Fgf8 mRNA associated with the posterior-to-anterior maturation of PSM cells[60]. Therefore, we suppose that this cell-autonomous decrease of Fgf/Erk signaling may act as a cell-intrinsic timer even in zebrafish embryos. Also, we cannot exclude the possibility that Fgf/Erk signaling may function upstream of the intrinsic timer. Further analysis of the molecular nature of cell-intrinsic timer and regulating mechanisms of Erk-dependent *ripply* expression should reveal this issue.

We also note that although it has already been proposed that the Tbx6 protein boundary coincides with the future somite boundary[14,16], this study shows for the first time that a boundary structure characteristic of somitic boundaries can be created at the boundaries of the Tbx6 protein domain (Fig. 1). Similar to somite boundaries, epithelization and fibronectin assembly occur along the boundary between *ripply*-deficient donor cells and *tbx6*-deficient host cells. Given that different gene expression occurs depending on the presence or absence of the transcriptional regulator Tbx6, such differences in gene expression along the boundary may lead to the generation of morphological boundaries. Since several *ephrin* genes are expressed in anterior PSM in dependence on *tbx6*[61], it seems reasonable to assume

that the Eph-ephrin interaction between the Tbx6 boundary plays an important role in generating the morphologically clear boundary[61,62].

In spite of gross similarity in somitogenesis, molecular mechanisms driving the segmentation clock are diverse among vertebrate species[39]. In mouse somitogenesis, *Ripply1* and *Ripply2* expression is activated by Mesp2[15,20], expression of which is cyclically induced by Notch signaling[21], which drives clock coupling with the Hairy-related transcriptional repressor. As a result, the segmentation clock cyclically activates *Ripply1* and *Ripply2* expression in the mouse. In contrast, previous studies showed that *mesps* and Notch signaling are dispensable for somite boundary definition in zebrafish[22,63]. Rather, we showed that Her1 and Her7 directly repress *ripply1* and *ripply2* expression in zebrafish (Supplementary Fig. 22). Of note, despite diversity in components of the segmentation clock and the role of Mesp between mouse and zebrafish embryos, the role of the Ripply-Tbx6 axis is highly conserved in the final process of somite boundary definition in mice and zebrafish[15,16]. This suggests that Ripply-mediated removal of Tbx6 proteins is likely to be one of the most fundamental processes of vertebrate somitogenesis.

## Methods
### Fish and embryos
Zebrafish with the TL2 background were used as the wild type, as described previously[64]. *tbx6*[til], *ripply1*[kt1032], *ripply2*[kt1034] and *Tg(her1:her1-Venus)*[bk15] have been described elsewhere[9,22,35]. Zebrafish were maintained at 28 °C under a 14-h light / 10-h dark cycle. Embryos were grown at 28.5 °C or 23.5 °C and their developmental stages were determined according to morphological criteria. The sex of embryos was not considered in this study because sex is determined after embryogenesis in zebrafish. This study was performed in accordance with the Guidelines for Animal Experimentation of the National Institutes of Natural Sciences, with the approval of the Animal Care and Use Committee (IACAC) of the National Institutes of Natural Sciences.

### Micro-injection of DNA, mRNA and sgRNA
Capped mRNA was synthesized by use of a message mega-machine kit (ThermoFisher; AM1340). The template for sgRNA was generated by PCR as previously described[65]. sgRNA was synthesized by incubation of templated DNA with 0.5 mM NTPs mix (Invitrogen; 18109017) and T7 RNA polymerase and purified using a Total RNA Extraction Column (FAVORGEN; FARBC-C50) with buffers for the RNeasy Micro Kit (QIAGEN; 74004). Plasmid DNA for injection was purified using PureLink™ HiPure Plasmid Midiprep Kit (Invitrogen™; K210004). Capped mRNA, DNA and/or sgRNA were dissolved in 0.2 M KCl with 0.05% Phenol Red and injected into one-cell zebrafish eggs using an IM300 microinjector (Narishige).

### Mutagenesis using TALEN
*her1*[kt1060] and *her7*[kt1061] mutants were generated using TALEN-mediated mutagenesis as described previously[22]. The target site was designed using TAL Effector Nucleotide Targeter 2.0 (https://tale-nt.cac.cornell.edu/node/add/talen-old). TALENs (left; NI NN NI NN NI NI NN NI NI NI HD NN NN NI NN NI NN NI HD, right; NG HD NN HD NI NN NG NG HD NG NG HD HD NI NI NI) and (left; NN NI NI HD HD NN NN NI NN NG HD NG NI NN NI NI NI NI HD, right; NG NN NG NN NG NG HD NG NN NN NN HD HD NG NG NN HD NI) were used for *her1*[kt1060] and *her7*[kt1061], respectively.

### Generation of transgenic fish
To generate *TG(msgn1;ggff)* fish, cDNA fragments encoding a fusion protein for the GAL4-DNA-binding domain and repeats of a minimal sequence of the VP16 transcription activator domain were amplified by PCR and cloned into sk+tol2 msgn1[66]. To generate *TG(hsp;her1)*, a ~3000-bp fragment of the *hsp70* promoter was amplified by PCR and cloned into pBS-Tol2B vectors[66] followed by cloning of the PCR-amplified *her1* cDNA fragment into the EcoRI/XbaI site. To generate

*TG(uas;ca-xmek1)*, a cDNA fragment encoding a ca-xmek1-CFP fusion protein was amplified by PCR and cloned into pT2AUASMCS (gifted from Kawakami Lab.). 25 pg of plasmid DNA were injected into one-cell embryos with 50 pg of *tol2* mRNA. F0 founders carrying mosaic integration of the transgene were mated with WT (TL2) fish and genotypes of F1 fish were identified by expression of a fluorescent signal during embryogenesis or PCR using genomic DNA extracted from clipped tails.

### Generation of V5 tag knock-in fish by ssODN-mediated genome editing
Capped Cas9 mRNA was synthesized using linearized pCS2 + hSpCas9 as a template, as previously described[67], followed by purification with a Total RNA Extraction Column (FAVORGEN; FARBC-C50) with buffers for the RNeasy Micro Kit (QIAGEN; 74004). 20 ng of ssODNs (ripply1-v5; TTGCCACTCCTCATACTAAAG<u>GCAAGCCTATCCCAAACCCTCTGC TGGGCCTGGACTCCA</u>CTATGGACAGTAAAATGCAG, ripply2-v5; CAC GAGCGGGTTAAACTCT<u>GGCAAGCCTATCCCAAACCCTCTGCTGGGCC TGGACTCCAC</u>AATGGACGCGAATCAACCCT (V5 encoding regions were underlined)) were injected into one-cell embryos with 100 pg of Cas9 mRNA and 50–100 pg of sgRNA, which recognize the sequence shown in Supplementary Fig. 9. Potential F0 founders were mated with WT (TL2) fish and genotypes of F1 fish were identified by PCR using genomic DNA extracted from clipped tails.

### Generation of *her7*[3xAchilles] KI fish
For knock-in vector construction, DNA fragments of 5′ homology arms with bait-D sequences[68] at 5′ ends and GS-linkers at 3′ ends carrying a silent mutation in the *her7*-sgRNA recognition site were amplified by PCR using primers: forward; 5′-ggggagctCCTCGCAGTCTAGGCCGAA GATCACCTGAAACTTCTGCTCCTG-3′, reverse; 5′-CCGgatCCGCCgCCT CCGGAgCCTCCGCCGCCGCTGCCTCCGCCTCCAGGCCAAGGTCTCCA GACAG-3′. DNA fragments for the 3′ homology arm with the bait-D sequence at 3′ end, FRT-2A-clover cassette and FRT-mscarlet-I were amplified by PCR using primers; 3′armF; 5′- ggggtaCCGgatCCGCC gCCTCCGGAgCCTCCGCCGCCGCTGCCTCCGCCTCCAGGCCAAGGTC TCCAGACAG −3′, 3′armR ggggtaCCTCGCAGTCTAGGCCGAAGATCG TGTGAGAACAGAAGTCGTGTGG, FRT-2A-F; 5′-GCGGatcCGGAGGCG GAGGCTCTgGAAGTTCCTATTCTCTAGAAAGTATAGGAACTTCtGGAA GCGGAGCTACAAACTTC-3′, FRT-2A-R; 5′-GAGAATAGGAACTTCcGC GGCCGCGAATTAAAAAACCTCC-3′, FRT-scl-F; 5′- CGCGGCCGCgGAA GTTCCTATTCTCTAGAAAGTATAGGAACTTCtGTGAGCAAGGGCGAGG CAGT-3′, FRT-scl-R; 5′-CAGAAGCttaCTTGTACAGCTCGTCCATGC-3′. These DNA fragments were sequentially assembled into pBS-SK+ vectors (pBS-her7-KI-FRT-2A-CLO-FRT-mscl). To generate DNA fragments containing triplet-repeated Achilles cDNA with an FRT sequence at the 5′ end, three DNA fragments of Achilles cDNA were amplified by PCR using primers; 1 F; 5′- ggggtaccGAATTCgGAAGTTCCTATTCTCT AGAAAGTATAGGAACTTCtGTGAGCAAGGGCGAGGAGCTGTTCAC-3′, 1R; 5′-gggcggccgcAAGCttaactcgagaaggatccCTTGTACAGCTCGTCCAT GC-3′, 2F; 5′-ggggatccATGGTGAGCAAGGGCGAGGA-3′, 2R; 5′-ggctcg agCTTGTACAGCTCGTCCATGC-3′, 3F; 5′- ggctcgagATGGTGAGCA AGGGCGAGGA-3′, 3R; 5′-ggAAGCttaCTTGTACAGCTCGTCCATGC-3′, and sequentially assembled into pBS-SK+ vectors (pBS-FRT-3xAchi). Finally, FRT-3xAchi fragment of pBS-FRT-3xAchi was substituted to CLO-FRT-mscl cassette of pBS-her7-KI-FRT-2A-CLO-FRT-mscl (pBS-her7-KI-FRT-2A-FRT-3xAchi). Then 25 pg of pBS-her7-KI-FRT-2A-FRT-3xAchi were injected with 100 pg of Cas9 mRNA and 50-150 pg of sgRNAs, which recognize; *her7*; CAGACTGTCTGGAGACCT<u>TGG</u> and BaitD; TCTTCGGCCTAGACTGCG<u>AGG</u>, into one-cell embryos. Candidate F0 founders were screened according to the fluorescent signal of Achilles in the muscle fiber at 24 hpf. F0 founders were crossed with WT (TL2) and knock-in fish were identified by the Achilles signal in paraxial mesoderm. Finally, a DNA fragment of the connection site was amplified by PCR using promoters; 5′F; 5′-AAGGATGAACCGGAG

TCTAG-3′, 5′R; 5′-GAATTCAGGTCCAGGGTTC-3′, 3′F; 5′-ACATGGTCCT GCTGGAGTTC-3′, 3′R; 5′-CGCATATTCGCACCCTTTAT-3′, and sequenced by direct PCR. The FRT-2A-FRT cassette was removed by injection of 100 pg of flp mRNA into one-cell embryos. Removal of the 2 A cassette was confirmed by PCR using primers; 5′- AAGGATGA ACCGGAGTCTAG-3′ and 5′-AGATCAGCTTCAGGGTCAGC-3′.

## Isolation of the *ripply2* promoter

A ~10-kbp fragment of *genomic* DNA containing the 8 kbp-*ripply2* upstream promoter was cut from BAC CH152C2 using BamHI and cloned into pBS-SK(+) vectors (pBS-r2-BamHI). Promoter activity was assessed by generation of transgenic fish carrying the isolated 8kb-*ripply2* promoter region connected to the cDNA fragment encoding *mclover2* and *ripply2* 3′UTR. To generate the plasmid construct for transgenesis, a *ripply2* 3′UTR fragment was amplified by PCR with primers; Forward; 5′-GGGTCGACAGCTTCTCCGCCAAAGCA-3′, Reverse; 5′-GGTCTAGACACCCCTCACAAGTCTACC-3′ and subcloned into SalI/XbaI site of pCS2SN vector (pCS2-r2utr). The DNA fragment containing the proximal region of the *ripply2* promoter and *mclover* cDNA was amplified by overlap extension PCR using primers; described below. Forward1; 5′- GGAATTCGCTAGCGCGAAGTCACCGTTTGT CAC-3′, Reverse1; 5′- CTCGCCCTTGCTCACCATAGTGTCCGTGGAAA GAG-3′, Forward2; 5′- CACGGACACTATGGTGAGCAAGGGCGAGGA-3′, Reverse2; 5′-GGGTCGACTCACTTGTACAGCTCGTCCATG-3′ and sub- cloned into the EcoRI/SalI site of pBS-SK+ vector (pBS-miniP-clo). The DNA fragment of *ripply2*-3′UTR and SV40pA was cut from pCS2-r2utr with SalI/KpnI and cloned into pBS-miniP-clo (pBS-miniP-clo-utr). Then a DNA fragment of the middle region of the *ripply2* promoter was cut from pBS-r2-BamHI with NheI/HindIII and subcloned into the 5′ region of pBS-miniP-clo-utr (pBS-r2-NheI- clo-utr). Then the distal part of the *ripply2* promoter was cut from pBS-r2-BamHI with BamHI/NheI and subcloned into the 5′ region of pBS-r2-NheI- clo-utr (pBS-r2-BamHII- clo-utr). Finally, the entire insert of pBS-r2-BamHII- clo-utr was cut out with NotI/AscI and cloned into the pBS-Tol2B vector (pT2- r2-BamHII- clo-utr). 25 pg of pT2- r2-BamHII- clo-utr plasmid were injected into one-cell embryos with 50 pg of tol2 transposase mRNA and promoter activity was confirmed according to *mclover* expression in F1 embryos.

## Plasmid construction for the reporter assay

A DNA fragment containing the proximal region of the *ripply2* pro- moter and the 3′ region firefly luciferase cDNA was amplified by overlap extension PCR with primers; Forward1; 5′- AAGTCACCGTTTGT CACGAG-3′, Reverse1; 5′-ATGTTTTTGGCGTCTTCCATAGTGTCCGTG GAAAGAGAGT-3′,Forward2;5′-ACTCTCTTTCCACGGACACTATGGAAG ACGCCAAAAACAT-3′, Reverse2; 5′-CACCTCGATATGTGCATCTG-3′, and subcloned into the HindIII/NarI site of pGL3-basic vector. Then, a DNA fragment of the middle and distal regions of the *ripply2* promoter was sequentially assembled in NheI/HindIII and BamHI/NheI site (pGL3-r2-bamH).

These mutations were introduced into the nine putative her1- binding sequences of the pGL3-r2-bamH vector (pGL3-r2-bamH-dN) by site-directed mutagenesis using overlap extension PCR with primers shown in Supplementary Table 2.

## Luciferase assay

To evaluate effects of Ripply1 and Ripply2 on *her1* promoter activity, pGL3-her1 was used as a reporter, as previously reported[32]. Details of expression vectors, pCS+ripply1, pCS+ripply2, pCS2 + her1 and pCXN2-m1CFP-xMEK1-SDSE have been described previously[23,32].

The day before transfection, $4 \times 10^4$ cells of a human embryonic kidney cell line (HEK293T cells) were plated into wells of a 24-well plate. Plasmids for reporter and expression vectors were transiently transfected using gene juice transfection reagent (Millipore;70967). pRL-TK vector was co-transfected as an internal control. Empty vectors were transfected to maintain a consistent amount of DNA in each

transfection. After the 24-h incubation, cells were harvested and luci- ferase activity was assessed using Dual-Glo™ Luciferase Assay System (Promega; E2920). All experiments were run in triplicate and sig- nificant differences were evaluated using one-way ANOVA, followed by the Tukey-Kramer test.

## Cell transplantation assay

2.5 ng of *tbx6* morpholino (5′-CATTTCCACACCCAGCATGTCTCGG-3′)[19] was injected into the one-cell stage of TL embryos to use as host embryos. For preparation of donor *ripply* dKOembryos, 2.5 ng of *rip- ply1* morpholino (5′-CATCGTCACTGTGTTTTTCGTTTTG-3′)[19] were injected into the one-cell stage of eggs obtained by crossing *rip- ply1^{kt1032/+}; ripply2^{kt1034/kt1034}* parents with 12 ng of rhodamine-dextran and 6 ng of biotin-dextran. Sphere to dome stage donor cells were trans- planted into the marginal zone of dome to 40% epiboly stage of host embryos. Embryos were fixed with 4% PFA at the 8 to 9-somite stage. Donor cells were visualized by incubation with 1/5000 diluted Rho- damine Red™-X conjugate of NeutrAvidin™ biotin-binding protein (Thermo Fisher; A6378) in PBS containing 1%DMOS, 0.1%TritonX100 and 2%BSA after IHC staining.

## Transient expression HA-Ripply1 expression assay

For construction of plasmids for transient expression of *ripply1* under control of the *hsp70l* promoter, DNA fragments of *nls-gfp-2A*, *HA-ripply1* and *ripply1* 3′UTR were amplified by PCR using primers; for *nls-gfp-2A*; forward; 5′-TGAAATCTAGTGGATCCGATAGCAAACATGCC AAAAAAGAAGAGAAAGGTAATGGTGAGCAAGGGCGAGGA-3′, reverse; 5′-AGGGTTCTCCTCCACGTCTCCAGCCTGCTTCAGCAGGCTGAAGTTT GTAGCTCCGCTTCCCTTGTACAGCTCGTCCATGC-3′, for *HA-ripply1*; forward;5′-CTGGAGACGTGGAGGAGAACCCTGGACCTTACCCATACG ATGTTCCAGATTACGCTAATTCTGTGTGCTTTGCCACTC-3′, reverse; 5′-TCAGTTGAAAGCTGTGAAGTGAC-3′, for *ripply1* 3′UTR; 5′-CACAACA GTCACTTCACAGC-3′, reverse; 5′-TTCTAGTCGAGGTCGACGATCACT GAAACCCTGCAAACCC-3′.

These three DNA fragments were connected by overlap extension PCR and subcloned into the pBS-Tol2B with 3000-bp fragment of *hsp70l* promoter.

25 pg of plasmid were injected early one-cell stage embryos obtained by crossing of *ripply1^{kt1032/+}; ripply2^{kt1034/kt1034}* parental fish with 50 pg of capped *tol2 TPase* mRNA. *ripply1^{kt1032}* and *ripply2^{kt1034}* double- homozygous embryos were selected by somite phenotype at the two- somite stage. Embryos were incubated at 38 °C for 30 min at 10.5 hpf after removal of the chorion. After heat shock treatment, embryos were incubated at 28.5 °C and fixed with 4% PFA at each time point for the IHC analysis.

## In situ hybridization

In situ hybridization was performed as previously described[69]. Digoxigenin-labeled RNA probes for *her1, her7, hes6, wnt8.a1, fgf8, msgn1, tbx6, ripply1* and *ripply2* were synthesized as previously described[20,23,32,66]. For detection of nascent mRNA, dechorionated embryos were fixed with a 2-h incubation at room temperature fol- lowed by an overnight incubation at 4 °C in 4% paraformaldehyde (PFA)/PBS. After fixation, embryos were dehydrated by treatment with MeOH and kept at −20 °C until use. Hybridization was performed at 55 °C and signals were detected using Anti-Digoxigenin-POD (Roche; 11207733910) and a TSA Plus Cyanine3/Fluorescein System (Perkin Elmer; 753001KT).

## Immunohistochemistry

For simultaneous detection of fibronectin and Tbx6, embryos fixed with 4%PFA were permeabilized by incubation in MeOH at −20 °C and then incubated with 1/200 diluted anti-fibronectin antibody (Sigma; F3648) and 1/200 diluted mouse anti-Tbx6 Antibody[40] overnight at 4 °C. Signals were detected with 1/700 diluted anti-rabbit IgG Alexa

488 (Thermo Fisher; A11008) and anti-mouse IgG Alexa 647 (Thermo Fisher; A21235), respectively. For simultaneous detection of fibronectin and F-Actin, embryos fixed with 4%PFA were permeabilized by treatment with PBS containing 0.5% Triton-X100 and then incubated with 1/200 diluted anti-fibronectin antibody overnight at 4 °C. Then, fibronectin and F-Actin signals were detected by incubation with 1/700 diluted anti-rabbit IgG Alexa 488 and 1/100 diluted Alexa Fluor™ 647 Phalloidin (Thermo Fisher; A22287). For simultaneous detection of GFP, HA-Ripply1 and Tbx6, embryos fixed with 4%PFA were permeabilized by incubation in MeOH at −20 °C and then incubated with 1/500 diluted chicken anti-GFP antibody (Abcam; ab13970), 1/500 diluted rat anti-HA antibody (3F10; Roche; 11867423001) and 1/666 diluted rabbit anti-Tbx6 Antibody[40] overnight at 4 °C. Signals were detected with 1/700 diluted anti-chicken IgG Alexa 488 (Thermo Fisher; A11039), anti-rat IgG Alexa 555 (Thermo Fisher; A21434) and, anti-rabbit IgG Alexa 647 (Thermo Fisher; A21245) respectively. For simultaneous detection of fibronectin and γ-Tubulin and β-Catenin, embryos fixed with 4%PFA were permeabilized by incubation in MeOH at −20 °C and then incubated with 1/1000 diluted mouseγ-Tubulin antibody (GTU-88; Sigma; T6557), 1/500 diluted rabbit anti-β-Catenin antibody (Sigma; C2206) overnight at 4 °C. Signals were detected with 1/700 diluted anti-mouse IgG Alexa 488 (Thermo Fisher; A11001) and, anti-rabbit IgG Alexa 647 (Thermo Fisher; A21245) respectively.

## Combination of Immunohistochemistry and in situ hybridization

For simultaneous detection of *ripply1* or *ripply2* nascent mRNA with Venus and dpERK protein, dechorionated embryos were fixed with a 2-h incubation at room temperature followed by an overnight incubation at 4 °C in 6%PFA/PBS with PhosSTOP™ phosphatase inhibitor cocktail (Merck; 4906845001). After the detection of nascent mRNA, embryos were incubated at 95 °C for 20 min in Antigen Unmasking Solution, Tris-Based (Vector; H-3301) containing 0.05% Tween20. After washing with PBSTw, embryos were incubated in blocking buffer (PBS containing heat-treated 5% fetal bovine serum, 2% BSA, 1% DMSO, and 0.1% Triton X-100) for 1 h at room temperature. Then, embryos were incubated with 1/2000 diluted anti-GFP Antibody (Thermo Fisher; A-11122), preabsorbed with a 1/10 volume of acetone powder extracted from zebrafish embryos, overnight at 4 °C. After 8x washes with PBSDTx (PBS with 0.1% Triton X-100), embryos were incubated with 1/2000 diluted HRP-conjugated anti-Rabbit IgG Antibody (Jackson Lab.111-035-114) overnight at 4 °C. After 8 washes with PBSDTx, signals were visualized with the TSA Plus Cyanine3/Fluorescein System. Then, embryos were incubated in MeOH/1%H₂O₂ for 30 min at room temperature. After 1 h blocking with blocking buffer, embryos were incubated with 1/2000 diluted anti-dpErk Antibody (Sigma; M9692) overnight at 4 °C. Then, embryos were washed with PBSDTx, incubated with 1/300 diluted HRP-labeled anti-mouse IgG Antibody (Promega; W402B) overnight at 4 °C. After 8 washes with PBSDTx, signals were detected with TSA-Cyanine5 (Perkin Elmer; SAT705A001KT) with buffers for the TSA Plus Cyanine3/Fluorescein System.

For simultaneous detection of V5-tagged Ripply and Tbx6 protein, embryos were incubated at 95 °C for 20 min in Antigen Unmasking Solution, Citrate-Based (Vector; H-3300) containing 0.05% Tween20. To detect V5-tagged Ripply, anti-V5 Antibody (Invitrogen; 46-0705) diluted to 1/2000 by Can Get Signal(R) immunostain Solution A (TOYOBO; NKB-501) was used for the first antibody reaction and signal was detected using 1/300 diluted HRP-labeled anti-mouse IgG Antibody (Promega) and the TSA Plus Cyanine3/Fluorescein System. Tbx6 protein was detected with 1/200 diluted mouse anti-Tbx6 Antibody[40] or 1/500 diluted rabbit anti-Tbx6 Antibody[16] with 1/800 diluted goat anti-mouse IgG Alexa 647 (Thermo Fisher; A21235) or anti-rabbit IgG Alexa 647(Thermo Fisher; A21245), respectively. After removal of the yolk, flat-mounts were imaged using a laser-scanning confocal microscope (SP8, Laica). Acquired images were analyzed using ImageJ (FIJI; 2.0.0).

## Genotyping

Genotypes of *ripply1ᵏᵗ¹⁰³²* and *ripply2ᵏᵗ¹⁰³⁴* were identified as previously reported[22]. Double-heterozygous fish of *her1ᵏᵗ¹⁰⁶⁰* and *her7ᵏᵗ¹⁰⁶¹* were identified with T7 endonuclease assay with primers; *her1* forward; 5′-GTACAACTTGCTCCGTCTAG-3′, reverse; 5′-CCTTGATCTCTCGCA GTCGC-3′, *her7* forward; 5′-CTATTGGAGTACACGTGCAATG-3′, reverse; 5′-TCCAGGATCTCTGCTTTCTC-3′. Genotypes of *her1;her7* double-homozygous fish were identified according to their phenotypes during somitogenesis. The *tbx6ᵗⁱˡ* genotype was assessed by detection of the mutation using direct PCR with primers: forward; 5′-TGCACTGCTAAAGCCTCATG-3′, reverse; 5′- CCTTCTTAGTGACGAT CATC-3′. *Tg(hsp;her1)* was genotyped by PCR using primers; forward; 5′- CCAGCGTTTGGAAGAACTGC-3′, reverse; 5′-TGGATCATGCGTGTC CTTGC-3′.

## Drug treatment of zebrafish embryos

For inhibition of Erk activity, PD184352 (abcam; ab141348) was dissolved in DMSO at 100 mM as a stock solution. Dechorionated embryos were incubated 10 μM PD PD184352 diluted in 1/3 Ringer's solution for 15 min at 28.5 °C.

## Time-lapse imaging

Dechorionated embryos were embedded in low-melting-point agarose dissolved in 1/3 Ringer's in fluorinated ethylene propylene tubes. Time-lapse images were acquired using light sheet microscopy (Z.1; Carl Zeiss) with a 1-min interval at 23.5 °C. Acquired images were analyzed using ImageJ(FIJI; 2.0.0).

## Statistical analysis

Differences between groups were examined using a Tukey-Kramer test or two-sided Wilcoxon rank sum test and Kruskal-Wallis test in R (The R Project for Statistical Computing). $p$ values < 0.05 were considered to indicate significant results.

## Mathematical models

We developed a mathematical model to examine whether the proposed gene regulatory network in Fig. 7a is sufficient for periodic Tbx6 boundary formation and segmentation clock arrest. To demonstrate sufficiency of the regulatory network, we chose a mechanistic model including dynamics of transcription factors such as Her, Ripply, and Tbx6 proteins. Here we describe details of the model.

For simplicity, we consider a one-dimensional domain $0 \le x \le L$ that describes the zebrafish PSM and somite region along the anterior-posterior axis of embryos (Supplementary Fig. 11a). We set $L = 400$ μm in simulations for wild type. In this one-dimensional domain, we arrange $N$ cells with a constant interval $\Delta x$. The distance between two neighboring cells $\Delta x$ represents cell size, and we set $\Delta x = 10$ μm. Then, the number of cells in the domain is $N = 41$ for the wild type. To represent the posterior growth of embryos, we chose the reference at the tailbud[70–72] (Supplementary Fig. 11b). In this tailbud reference frame, $x = L = (N-1)\Delta x$ is the posterior tip of the tailbud and the domain describes the tissue anterior to the tailbud until $x = 0$ μm. Note that we do not set the position of the anterior end of the PSM a priori. It is determined by interactions between transcription factors in the model. $x = x_i(t)$ is the location of cell $i$ ($i = 0,1,2,\ldots$) at time $t$. We assume that due to posterior progression of the tailbud caused by axis extension of embryos, cells move anteriorly relative to the tailbud (Supplementary Fig. 11b). This relative motion of cells is termed cell advection. We denote the constant advection speed of these cells as $u$. Then, the relative position of cell $i$ changes as $x_i(t) = L - u \cdot (t - t_i)$ for $t > t_i$ where $t_i$ is the time at which the cell $i$ enters to the domain at $x_i(t_i) = L$. Cell $i$ reaches $x = 0$ at $t = t_i + L/u$. Then, we remove this cell, and add a new cell at the posterior end $x = L$ to keep the cell number $N$ in the domain constant. We copy the concentration variables of the

nearest left neighbor to the newly added cell. For the illustration of mRNA and protein expression patterns in figures, we convert the tailbud reference frame into the lab reference frame for ease of interpretation (Supplementary Fig. 11b).

We describe time evolution of concentrations of mRNAs and proteins using delay differential equations based on previous modeling work[41,45–47]. In zebrafish, there are *her1* and *her7* genes, and *ripply1* and *ripply2* genes. In the current description, we only consider one of these homologous genes for simplicity, and refer to them as *her* and *ripply*, respectively. Also, we do not consider intercellular interaction through Delta-Notch signaling in order to avoid additional complexity. The model includes concentrations of nascent and mature *her* and *ripply* mRNAs to directly compare simulation results with corresponding in situ hybridization data. Nascent mRNAs include both introns and exons, whereas mature mRNAs include only exons. We denote the concentrations of nascent and mature *her* mRNAs as $m_h$ and $M_h$, and those of nascent and mature *ripply* mRNAs as $m_r$ and $M_r$, respectively. The model also includes concentrations of Her, Ripply, Tbx6, and dpErk proteins as variables, $p_h$, $p_r$, $p_t$, and $p_e$, respectively. We normalize concentrations of mRNAs by the ratio of the maximum transcription rate to the degradation rate of *her* mRNA. Similarly, we normalize concentrations of proteins by the ratio of the production rate of Tbx6 protein by the positive feedback loop to its degradation rate. Hence, all concentration variables in the equations are dimensionless.

We describe the time evolution of *her* mRNAs for cell $i$ as:

$$\frac{1}{\mu_1}\frac{dm_h^{(i)}(t)}{dt} = \frac{1}{1+\left(p_h^{(i)}(t-\tau_h)/K_1\right)^{n_1}} \cdot f\left(p_t^{(i)}(t-\tau_h)\right) - \mu_{sh} \cdot m_h^{(i)}(t), \quad (1a)$$

$$f(y) = \begin{cases} \frac{(y/K_2)^{n_2}}{1+(y/K_2)^{n_2}} & \text{for } x_i \leq x_h \\ 1 & \text{for } x_i > x_h \end{cases}, \quad (1b)$$

$$\frac{1}{\mu_1}\frac{dM_h^{(i)}(t)}{dt} = \mu_{sh} \cdot m_h^{(i)}(t) - M_h^{(i)}(t). \quad (1c)$$

The first term of Eq. (1a) represents transcription of *her* mRNA. Transcription is repressed by Her protein $p_h$, whereas it is induced by Tbx6 protein $p_t$. *her* expression is observed in the posterior PSM where expression of Tbx6 protein is absent. In contrast, in the anterior PSM, our experimental data indicate that Tbx6 protein is required for *her* expression. Therefore, we assume that the sensitivity of transcription of *her* mRNA to Tbx6 protein changes at $x_h$ as described in Eq. (1b). $\tau_h$ is the time required for transcribing *her* mRNA. The second term of Eq. (1a) and the first term of Eq. (1c) are the splicing of nascent (*her-intron*) mRNA. The second term of Eq. (1c) is the degradation of mature (*her-exon*) mRNA. We set the degradation rate of *her-exon* mRNA $\mu_1$ as the reference timescale of mRNAs.

Similarly, we model the time evolution of concentrations of nascent and mature *ripply* mRNAs for cell $i$ as:

$$\frac{1}{\mu_1}\frac{dm_r^{(i)}(t)}{dt} = \nu_1 \frac{1}{1+\left(p_h^{(i)}(t-\tau_r)/K_3\right)^{n_3}} \frac{1}{1+\left(p_e^{(i)}(t-\tau_r)/K_4\right)^{n_4}}$$
$$\times \frac{\left(p_t^{(i)}(t-\tau_r)/K_5\right)^{n_5}}{1+\left(p_t^{(i)}(t-\tau_r)/K_5\right)^{n_5}} - \mu_{sr}m_r^{(i)}(t), \quad (2a)$$

$$\frac{1}{\mu_1}\frac{dM_r^{(i)}(t)}{dt} = \mu_{sr}m_r^{(i)}(t) - \mu_2 M_r^{(i)}(t). \quad (2b)$$

The first term of Eq. (2a) represents transcriptional regulation of *ripply* mRNA. Her protein $p_h$ and dpErk protein $p_e$ repress transcription, whereas Tbx6 protein $p_t$ induces it. $\tau_r$ is the time needed to transcribe *ripply* mRNA. Nascent *ripply-intron* mRNA is spliced at the rate $\mu_{sr}$. Mature mRNA (*ripply-exon*) is degraded at the rate $\mu_2$.

Translation of mature mRNAs produces Her and Ripply proteins. The time evolution of their protein levels for cell $i$ is described as:

$$\frac{1}{\mu_3}\frac{dp_h^{(i)}(t)}{dt} = \nu_2 M_h^{(i)}\left(t-\tau_H(x_i)\right) - \mu_4 p_h^{(i)}(t), \quad (3a)$$

$$\tau_H(x) = \tau_{H0} + \frac{\tau_{HL}-\tau_{H0}}{L}x, \quad (3b)$$

$$\frac{1}{\mu_3}\frac{dp_r^{(i)}(t)}{dt} = \nu_3 M_r^{(i)}(t-\tau_R) - \mu_5 p_r^{(i)}(t). \quad (4)$$

The first terms of Eqs. (3a) and (4) represent translation, and the second terms are the degradation of the proteins. $\tau_H(x)$ and $\tau_R$ describe time delays that are caused by translation, transport of proteins from cytoplasm to nucleus, and protein modifications. We further assume that the time delay of Her protein $\tau_H(x)$ is a function of the space (Supplementary Fig. 11c[46]). For simplicity, we assume a linear gradient for $\tau_H(x)$ as in Eq. (3b). $\tau_{H0}$ and $\tau_{HL}$ are the delay values at $x=0$ and $L$, respectively. In simulations, the spatial gradient of $\tau_H$ generates a gradient of oscillation period along the anterior-posterior axis, thereby causing traveling waves of *her* expression. (Supplementary Fig. 12). In ref. 46, translational time delays of Her1 protein across the PSM were estimated from spatial distances between stripes of *her1* mRNA and Her protein in *her1*-Venus transgenic embryos at somite stages 12-14. This previous study[46] reported that translational time delays increased 4.4-fold from the intermediate to the anterior PSM. Based on this estimate, we assumed that the ratio of translational delays between the anterior and posterior ends $\tau_{H0}/\tau_{HL}$ is in the range $1 \leq \tau_{H0}/\tau_{HL} \leq 7$. In Fig. 7 in the main text, we set $\tau_{H0}/\tau_{HL}=6$.

Because we do not measure *tbx6* mRNA in our experiment, we only consider Tbx6 protein in the model to reduce the number of variables. We describe the time evolution of Tbx6 protein levels $p_t$ for cell $i$ as:

$$\frac{1}{\mu_3}\frac{dp_t^{(i)}(t)}{dt} = \frac{\left(p_t^{(i)}(t)/K_6\right)^{n_6}}{1+\left(p_t^{(i)}(t)/K_6\right)^{n_6}} + \gamma \frac{1}{1+\left(F(x_i)/K_{FR}\right)^{n_R}} \frac{\left(F(x_i)/K_{FA}\right)^{n_A}}{1+\left(F(x_i)/K_{FA}\right)^{n_A}}$$
$$- \left(1+\eta p_r^{(i)}(t)\right)p_t^{(i)}(t). \quad (5a)$$

The first term represents positive feedback regulation of Tbx6 protein[44]. $K_6$ is the dissociation constant of Tbx6 protein for its own promoter and a small $K_6$ value represents strong positive feedback regulation. The second term is activation of Tbx6 production by a posterior signaling gradient $F(x)$, such as Fgf/Erk. To reproduce the spatial expression pattern of Tbx6 protein in the posterior PSM, we assume an incoherent feedforward loop (IFFL)[48]. One of the two pathways activates Tbx6 production, whereas the other represses it (Supplementary Fig. 11d). For simplicity, we omit the intermediate repressor in Eq. (5a) so that signal $F$ both activates and represses production of Tbx6. Because we assume that the dissociation constant $K_{FA}$ is smaller than $K_{FR}$ (Supplementary Table 1), activation of Tbx6 is allowed only in spatial positions with a medium range of signal intensity $F(x)$. For simplicity, we do not consider time delays in Tbx6 production. The third term in Eq. (5a) is the degradation of Tbx6 protein. Degradation of Tbx6 is increased proportional to Ripply protein levels with coefficient $\eta$. We also consider the basal degradation of Tbx6 protein that is independent of Ripply protein. Its

degradation rate $\mu_3$ is the reference time scale for the dynamics of protein concentrations. Based on Eq. (5a), we define

$$P \equiv \frac{(p_t(x)/K_6)^{n_6}}{1+(p_t(x)/K_6)^{n_6}} + \gamma \frac{1}{1+(F(x)/K_{FR})^{n_R}} \frac{(F(x)/K_{FA})^{n_A}}{1+(F(x)/K_{FA})^{n_A}},$$

and

$$D \equiv (1+\eta p_r(x))p_t(x),$$

in Fig. 7c in the main text. $1+\eta p_r$ is the change of Tbx6 degradation rate by Ripply. In Fig. 7d and Supplementary Fig. 16, we used the value of $1+\eta p_r$ with a peak value of $p_r$ around the anterior border of dpErk. The condition $P=D$ provides the steady states of Tbx6 protein $dp_t/dt = 0$.

We describe the signaling gradient $F(x)$ in Eq. (5a) as:

$$F(x) = F_0 \exp\left(-\frac{L-x}{x_f}\right), \tag{5b}$$

where $F_0$ is the signal intensity at the tailbud tip $x=L$, and $x_f$ is the length scale of the signaling gradient. We assume that the signaling gradient moves together with the tailbud. Hence, the shape of the gradient remains unchanged in the tailbud reference frame.

In this way, we assume an IFFL with the posterior gradient of Fgf/Erk signaling as an input to *tbx6*. The combination of PFL and IFFL can generate the posterior border of the Tbx6 expression domain in the following way (Supplementary Fig. 12). In the model, new cells are added to the tailbud with low expression levels of Tbx6. As cells leave the tailbud, Tbx6 expression is activated by $F$ through the IFFL. Due to this activation, only a steady state with high Tbx6 levels remains stable and cellular states converge to this stable steady state. As these cells reach the anterior PSM, they lose activation by $F$, leading to the reappearance of the stable steady state with low Tbx6 levels. The presence of such IFFL in zebrafish is not yet known, but in *Drosophila* embryos, combinations of maternal factor gradients and FFLs composed of segmentation genes produce their striped expression patterns along the anterior-posterior axis[49]. Hence, a similar FFL may be responsible for regulation of the Tbx6 expression pattern in zebrafish.

Finally, we describe the spatial distribution of dpErk as:

$$p_e(x,t) = \begin{cases} 0 & \text{for } x < x_e(t) \\ \nu_e\left(1 - e^{-q(x-x_e(t))}\right) & \text{for } x \geq x_e(t) \end{cases}, \tag{6}$$

where $x_e(t)$ denotes the anterior border of dpErk expression. The assumption that it is fixed at the position $x_e(t)=x_E$ in the tailbud reference frame means that the dpErk pattern moves together with the tailbud. $x_e(t)$ may change over time to describe the movement of the dpErk gradient relative to the tailbud. Indeed, we validate whether a sharp border of Tbx6 expression domain is formed by Ripply in the presence of stepwise posterior shift of dpErk gradient[52–54]. To model this, we assume that $x_e(t)=x_E - ut$ for $x_e(t) > x_E - \delta$, meaning that the boundary position of dpErk expression is fixed in the lab reference frame. Then, posterior shift of the dpErk pattern occurs as $x_e(t) \leftarrow x_E$ when $x_e(t)=x_E - \delta$. Thus, $\delta$ is the length that the dpErk gradient travels at each stepwise shift. In simulations for the dpErk stepwise shift, we set $x_E = 200\ \mu m$ and $\delta = 60\ \mu m$ in the tailbud reference frame. The initial position of the dpErk boundary was $x_e(0)=180\ \mu m$.

The initial history of numerical simulations is the following. In all cells in the array, we set $m_h(t)=1, m_r(t)=0, M_h(t)=1, M_r(t)=0, p_h(t)=\nu_2/\mu_4$, and $p_r(t)=0$ for $-\max(\tau_h,\tau_r,\tau_{H0},\tau_R) \leq t \leq 0$. The initial history of Tbx6 protein depends on a cell's position, $p_t(t)=1$ $(x \leq x_h)$ and $p_t(t)=0$ $(x > x_h)$. For dpErk, we used Eq. (6) to set the initial history. We integrated the above delay differential equations until $t = 460$ min

to obtain a regular spatio-temporal pattern. Then, we shifted time ($t \leftarrow 0$ min) and plotted these patterns as shown in the main text and Supplementary Figures. Delay differential equations were numerically solved by the Euler method with a time step of 0.01 min. The custom simulation code was written in C language and is provided as Supplementary Software 1. Values of parameters for wild type simulation are listed in Supplementary Table 1.

We model the *her* knockout (KO) mutant by multiplying the transcription term in Eq. (1a) by zero. Similarly, the *ripply* KO mutant is modeled by setting $\nu_1 = 0$ in Eq. (2a). The dpErk knock-down experiment by a MEK1 inhibitor is modeled by reducing the value of $\nu_e$ in Eq. (6) by ten-fold from that of wild type.

For the bifurcation analysis of *her* oscillation with respect to Tbx6 protein levels (Supplementary Fig. 17), we use Eqs. (1) and (3), which describe behaviors of *her* gene products for a single cell. In Eq. (1), we consider constant values of Tbx6 protein levels as a bifurcation parameter. We also use a constant value of translational delay in Eq. (3a), $\tau_H = 7.45$ min that is the delay value near the anterior border of Tbx6 protein expression domain.

### Reporting summary

Further information on research design is available in the Nature Portfolio Reporting Summary linked to this article.

### Data availability

Raw data associated with the figures are included in the Source Data file. All the other data are available within the article and its Supplementary Information. Source data are provided with this paper.

### Code availability

The C program and Mathematica codes that were used to simulate the mathematical model and visualize results, respectively are provided with this paper as Supplementary Software 1.

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

## Acknowledgements

The authors thank the Spectrography and Bioimaging Facility of the NIBB Core Research Facilities for their technical support. We also thank S. Amacher, A.C. Oates, K. Kawakami, K. Hoshijima, Y. Niino, A. Miyawaki, K. Aoki, M.Kinoshita, S. Ansai for providing materials and all members of ST's laboratory are gratefully acknowledged for helpful discussions. This work was supported by the following programs: Grants-in-Aid for Scientific Research (B), 18H02454 and 21H02498 to S.T., Grants-in-aid for Scientific Research on Innovative Areas, 24111002, 17H05782, 19H04797 to S.T. and 17H05762, 19H04772 to K.U., Grants-in-Aid for Scientific Research (C), 17K07423 to TY and 20K06653 to K.U., Grants-in-aid for Transformative Research Areas, 22H05642 to S.T., from the Japan Society for the Promotion of Science. Additional support came from a grant from the Exploratory Research Center on Life and Living Systems (ExCELLS) to T.Y. and K.U.

## Author contributions

T.Y. planned and performed all experiments and wrote the text. K.U. formulated the mathematical concept, conducted computer modeling, and wrote the text. S.T. participated in planning of experiments and wrote the text. All authors reviewed and approved the manuscript.

## Competing interests

The authors declare no competing interests.
