## [Peer Review File · Nature Communications]

Tbx6/Ripply mechanism as a dynamic to static converter in somite segmentationEditorial Note: This manuscript has been previously reviewed at another journal that is not operating a transparent peer review scheme. This document only contains reviewer comments and rebuttal letters for versions considered at *Nature Communications* .

REVIEWERS' COMMENTS

Reviewer #1 (Remarks to the Author):

The authors have satisfactorily addressed my previous comments, so I am very supportive of this interesting study.

Reviewer #2 (Remarks to the Author):

Yabe et al., A Tbx6/Ripply mechanism as dynamic-to-static converter in somite formation

In this manuscript, the authors investigate the mechanism for stopping the travelling segmentation clock wave and the establishment of the anterior border of Tbx6 protein expression in zebrafish. The authors detail the positive and negative regulatory interactions between several genes involved in the segmentation clock and the formation of somite boundaries, namely Ripply1 and 2, Tbx6, Her1 and 7, and Erk. In the course of establishing these relationships, they strikingly demonstrate extended propagation of the segmentation clock beyond the anterior presomitic mesoderm in Ripply1,2 double knockouts. The authors assemble the investigated genes into a gene regulatory network that combines bistable Tbx6 protein expression (controlled by cross-repressive interaction between Ripply and Her) with a wavefront of Erk activity. This regulatory network is then used to construct a mathematical model that successfully describes the generation of successive Tbx6 anterior expression borders and the stop of the segmentation clock wave. New to this version of the manuscript is the demonstration that a border between Tbx6-high and Tbx6-low cells is sufficient for somite boundary formation.

The authors have responded well to prior comments. But this new version of the manuscript largely remains a refinement of the group's previous work. However, the additional evidence that the anterior border of Tbx6 expression is directly responsible for physical boundary formation synergizes with the paper's goal to identify a minimal network of genes responsible for somite boundary determination. With the new finding, the paper works as a collated description of the model of somite boundary formation in zebrafish.

Other Comments

1. Figure 1 presents a striking result. However, the boundary should be further characterized and quantified. The cells at the ectopic boundary appear to be arranged relative to it but the authors should demonstrate whether they take on epithelial features such as cell morphology and/or nuclear position.

2. Due to the importance of the Tbx6 expression border's role in boundary formation, greater emphasis should be placed on the new finding in the paper's Discussion.

Minor comments

1. Extending the leftmost dashed line in fig 7b to the bottom panel would clarify the relationship of the diagram to successive somite boundary formation.

Reviewer #3 (Remarks to the Author):

The authors addressed most of my points and I am by and large happy with the revision and arguments presented, with the exception of two points:

- I am still not convinced by the author's conclusion that "mathematical simulation based on the regulatory network identified in this study clearly show that the segmentation clock controls somite boundaries without fixation of its own phase" . If I understand correctly, in their reply the authors argue that, because $her1$ is not fixed, there is no fixation of the clock phase. This is in my opinion a misleading answer based on a narrow definition of fixation of the phase : we know there are R-C markers in the somite, which likely are regulated by clock genes, and for this reason very likely to fix in some way the clock phase. In other words, clock fixation might not be done by clock genes, and I do not think the authors can tell that somite boundaries definition happens without fixation of the phase. Maybe the authors imply that their model or data (based on the output of the clock in the anterior) do not suggest it, but since they are not examining such possible R-C marker dynamics again I do not think they can formally exclude this scenario.
- the authors mention that the period is multiplied by 1.3 only between posterior and anterior, and use it to exclude some bifurcation scenarios. But Shih et al only looked at the last few cycles, multiple papers (e.g. by Julian Lewis, see e.g. Giudicelli 2007) have argued that the period is increasing much more than that (at least 3 fold), in the more extensive study of Rohde 2021, in Fig. 1B the increase in period is again more like a factor 3 at least, with furthermore a clear non-linear effect as cells get more anterior, very consistent with the older measurements based on in-situs. It is also likely that most measurements based on peak to peak distance underestimate the period increase since one would not know what happens after the last peak. Without more careful discussions of such data I would refrain from formally excluding bifurcation scenarios from the model (it is difficult to explain such increases with a standard Hopf model, without engineering somehow by hand a period increase, e.g. with increasing phenomenological "delays" as done here). To be clear, I do not think the precise nature of bifurcations matter in any way in the authors' model as soon as there is a moderate period gradient, so I do not think too much focus is required on the nature of bifurcations inferred from the model. But if the authors really want to discuss bifurcations, then they should address discrepancies with those data.

In the following text, our responses to the comments are shown by blue Arial font.

Our response to the suggestions proposed by the reviewer #1

The authors have satisfactorily addressed my previous comments, so I am very supportive of this interesting study.

-----We would like to express our sincere gratitude to this reviewer for his/her very helpful comments so far.

Our response to the suggestions proposed by the reviewer #2

Yabe et al., A Tbx6/Ripply mechanism as dynamic-to-static converter in somite formation

In this manuscript, the authors investigate the mechanism for stopping the travelling segmentation clock wave and the establishment of the anterior border of Tbx6 protein expression in zebrafish. The authors detail the positive and negative regulatory interactions between several genes involved in the segmentation clock and the formation of somite boundaries, namely Ripply1 and 2, Tbx6, Her1 and 7, and Erk. In the course of establishing these relationships, they strikingly demonstrate extended propagation of the segmentation clock beyond the anterior presomitic mesoderm in Ripply1,2 double knockouts. The authors assemble the investigated genes into a gene regulatory network that combines bistable Tbx6 protein expression (controlled by cross-repressive interaction between Ripply and Her) with a wavefront of Erk activity. This regulatory network is then used to construct a mathematical model that successfully describes the generation of successive Tbx6 anterior expression borders and the stop of the segmentation clock wave. New to this version of the manuscript is the demonstration that a border between Tbx6-high and Tbx6-low cells is sufficient for somite boundary formation.

The authors have responded well to prior comments. But this new version of the manuscript largely remains a refinement of the group's previous work. However, the additional evidence that the anterior border of Tbx6 expression is directly responsible for physical boundary formation synergizes with the paper's goal to identify a minimal network of genes responsible for somite boundary determination. With the new finding, the paper works as a collated

description of the model of somite boundary formation in zebrafish.

-----First of all, we would like to thank this reviewer for appreciating our response.

Other Comments

1. Figure 1 presents a striking result. However, the boundary should be further characterized and quantified. The cells at the ectopic boundary appear to be arranged relative to it but the authors should demonstrate whether they take on epithelial features such as cell morphology and/or nuclear position.

-----We also think this point is important. To confirm that the cells lined up along the ectopic Tbx6 expression boundary are actually epithelialized, we investigated whether the centrosome, which is known to be located on the apical side of epithelial cells, is really localized in this way. As shown in Supplementary Fig. 2, the centrosome, which was randomly located in the control, was located almost opposite the boundary of Tbx6 expression, indicating that the cell is actually epithelialized with the boundary side as the basal side, as in normal somites.

2. Due to the importance of the Tbx6 expression border's role in boundary formation, greater emphasis should be placed on the new finding in the paper's Discussion.

-----We understand the importance of this finding. However, due to limitation of word number, we decided to describe this finding only in the Result section. We will discuss this point precisely in some review article in future.

Minor comments

1. Extending the leftmost dashed line in fig 7b to the bottom panel would clarify the relationship of the diagram to successive somite boundary formation.

-----We corrected this figure according to this suggestion.

Our response to the suggestions proposed by the reviewer #3

The authors addressed most of my points and I am by and large happy with the revision and arguments presented, with the exception of two points:

-----We would like to thank this reviewer for appreciating our response.

- I am still not convinced by the author's conclusion that "mathematical simulation based on the regulatory network identified in this study clearly show that the segmentation clock controls somite boundaries without fixation of its own phase" . If I understand correctly, in their reply the authors argue that, because *her1* is not fixed, there is no fixation of the clock phase. This is in my opinion a misleading answer based on a narrow definition of fixation of the phase : we know there are R-C markers in the somite, which likely are regulated by clock genes, and for this reason very likely to fix in some way the clock phase. In other words, clock fixation might not be done by clock genes, and I do not think the authors can tell that somite boundaries definition happens without fixation of the phase. Maybe the authors imply that their model or data (based on the output of the clock in the anterior) do not suggest it, but since they are not examining such possible R-C marker dynamics again I do not think they can formally exclude this scenario.

----In our previous manuscript, we defined "fixation of the clock" as the sustained expression of a clock gene without moving like a wave while retaining phase information. On the other hand, this reviewer seems to propose that "fixation of clock" can be considered even in the case where the phase information of clock gene expression is transmitted to downstream genes and subsequently determines the rostro-caudal polarity of somites. To avoid confusion, we modified the relevant part as follows.

(Previous) "mathematical simulation based on the regulatory network identified in this study clearly show that the segmentation clock controls somite boundaries without fixation of its own phase"

(Revised) "mathematical simulation based on the regulatory network identified in this study clearly show that the segmentation clock controls somite boundaries without fixation of expression waves in space."

- the authors mention that the period is multiplied by 1.3 only between posterior and anterior, and use it to exclude some bifurcation scenarios. But Shih et al only looked at the last few cycles, multiple papers (e.g. by Julian Lewis, see e.g. Giudicelli 2007) have argued that the period is increasing much more than that (at least 3 fold), in the more extensive study of Rohde 2021, in Fig. 1B the increase in period is again more like a factor 3 at least, with furthermore a clear non-linear effect as cells get more anterior, very consistent with the older measurements based on in-situ. It is also likely that most measurements based on peak to peak distance underestimate the period increase since one would not know what happens after the last peak. Without more careful discussions of such data I would refrain from formally excluding bifurcation scenarios from the model (it is difficult to explain such increases with a standard Hopf model, without engineering somehow by hand a period increase, e.g. with increasing

phenomenological "delays" as done here). To be clear, I do not think the precise nature of bifurcations matter in any way in the authors' model as soon as there is a moderate period gradient, so I do not think too much focus is required on the nature of bifurcations inferred from the model. But if the authors really want to discuss bifurcations, then they should address discrepancies with those data.

-----We are afraid that our reply in the previous revision might cause misinterpretation. We did not mean to argue against the possibility of an infinite period bifurcation in living vertebrate embryos. Rather, we tried to clarify that, in our mathematical model, oscillation is arrested through a Hopf bifurcation. As the reviewer pointed out, to reveal the bifurcation scenarios for oscillation arrest in embryos, one needs to understand the molecular mechanism for how the period of oscillation increases as cells are translocated from the posterior to anterior PSM. The regulation of oscillation period across the PSM is an important question, but this is not the main scope of the current paper. Therefore, we chose to refrain from the detailed discussion of the nature of bifurcation.

Following the reviewer's advice, we removed the term "Hopf bifurcation" from the revised version of the main text (page 11, line 327-328). In addition, we rephrased a sentence about the stop of Her1 expression in discussion to avoid giving impression that we conclude that the oscillation arrest is caused by Hopf bifurcation in zebrafish embryos (page 13, line 379). Still, we would like to retain Supplementary Fig. 17 for the completeness of the mathematical analysis and for readers who might be interested in how the oscillation is arrested in our mathematical model. We rephrased the legend of Supplementary Fig. 17 according to the changes in the main text.